# BANZ-FS: BANZSL FINGERSPELLING DATASET

**Xin Shen**[1], **Yan Ke**[1], **Xinyu Wang**[1], **Xin Yu**[2*]

[1]The University of Queensland

[2]Australian Institute for Machine Learning, Adelaide University

x.shen3@uqconnect.edu.au

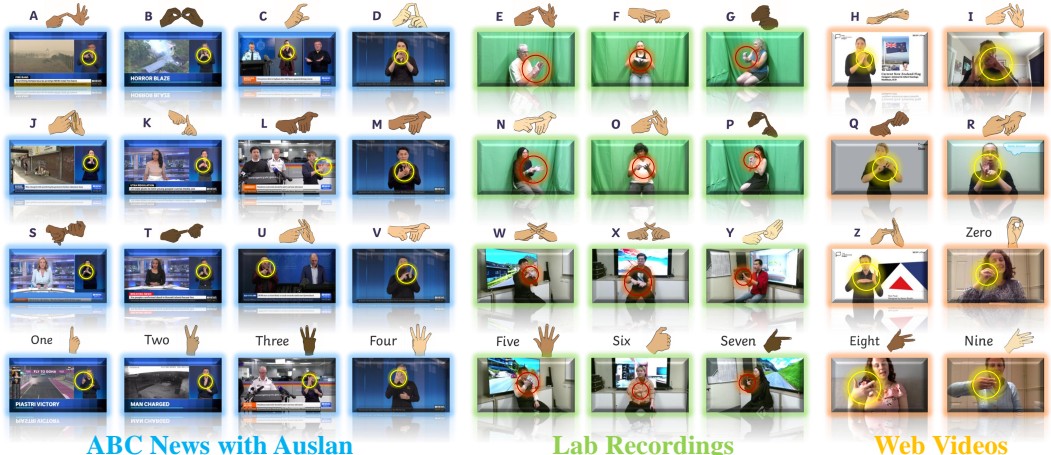

Figure 1: **Overview of BANZ-FS dataset sources and coverage.** The figure shows fingerspelling instances for all 26 letters (A–Z) and 10 digits (0–9) from three sources: ABC News, Lab Recordings, and Web Videos. News instances reflect formal, live interpretations by professional signers. Lab recordings offer clean, controlled settings ideal for analysis. Web videos capture diverse, in-the-wild signing styles across various environments.

## ABSTRACT

Fingerspelling plays a vital role in sign languages, particularly for conveying names, technical terms, and words not found in the standard lexicon. However, evaluation of *two-handed* fingerspelling detection and recognition is rarely addressed in existing sign language datasets—particularly for **BANZSL** (**B**ritish, **A**ustralian, and **N**ew **Z**ealand **S**ign **L**anguage), which share a common two-handed manual alphabet. To bridge this gap, we curate a large-scale dataset, dubbed **BANZ-FS**, focused on BANZSL fingerspelling in both controlled and real-world environments. Our dataset is compiled from three distinct sources: (1) live sign language interpretation in news broadcasts, (2) controlled laboratory recordings, and (3) diary vlogs from online platforms and social media. This composition enables BANZ-FS to capture variations in signing tempos and fluency across diverse signers and contents. Each instance in BANZ-FS is carefully annotated with multi-level alignment: video ↔ subtitles, video ↔ fingerspelled letters, and video ↔ target lexicons. In total, BANZ-FS includes over 35,000 video-aligned fingerspelling instances. Importantly, BANZ-FS highlights the unique linguistic and visual challenges posed by two-handed fingerspelling, including handshape coarticulation, self-occlusion, intra-letter variation, and rapid inter-letter transitions. We benchmark state-of-the-art models on the key tasks, including fingerspelling detection, isolated fingerspelling recognition, and fingerspelling recognition in context. Experimental results show that BANZ-FS presents substantial challenges while offering rich opportunities for BANZSL understanding and broader sign language technology. The dataset and benchmarks are available at ⌂ BANZ-FS.

---

*Corresponding author.

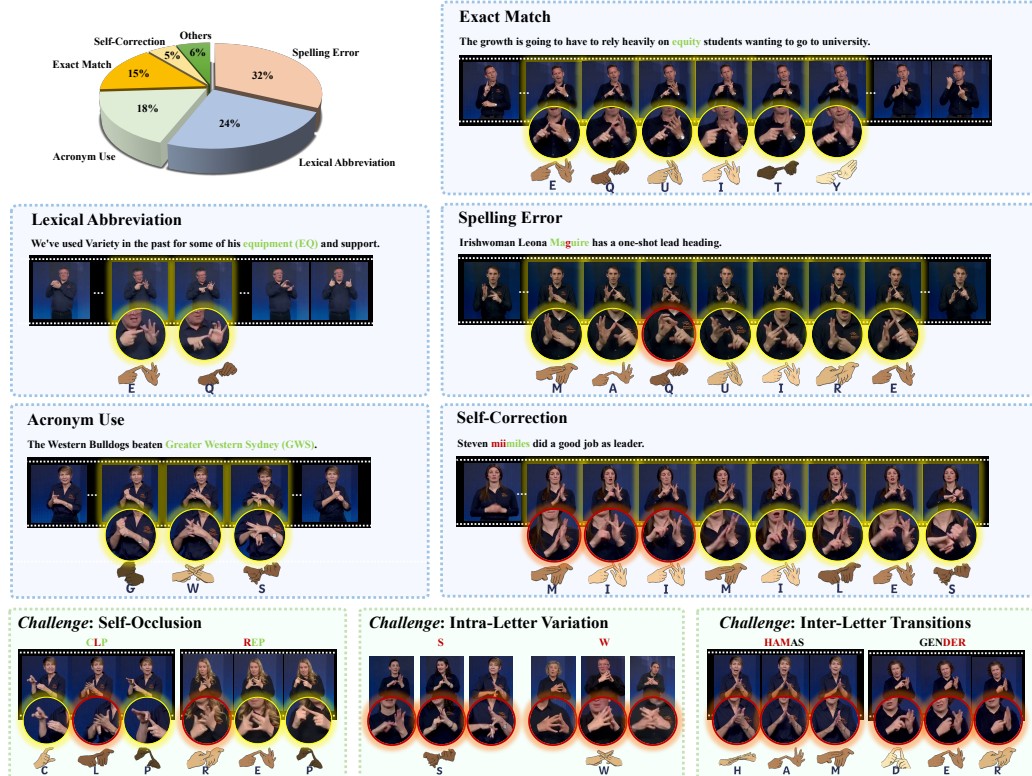

Figure 2: **Overview of typical fingerspelling phenomena and visual challenges captured by the BANZ-FS dataset.** The pie chart (top-left) illustrates the proportion of different fingerspelling phenomena annotated within the dataset. Representative examples below highlight diverse real-world cases, such as exact matches ("equity"), lexical abbreviations ("equipment" → "EQ"), spelling errors ("Maguire" misspelled as "Maquire"), acronym use ("Greater Western Sydney" → "GWS"), and inline corrections ("miimiles" corrected to "miles"). The bottom row (green boxes) highlights key visual challenges specific to two-handed fingerspelling systems, such as self-occlusion, intra-letter variation, and rapid inter-letter transitions, further underscoring the complexity of accurate fingerspelling recognition and translation in BANZSL.

# 1 INTRODUCTION

Sign languages (SL) are natural languages that serve as primary modes of communication for Deaf and hard-of-hearing individuals, enabling rich self-expression and full participation in society. Like spoken languages, sign languages possess their own grammars and lexicons, and they vary widely across regions—even in places that share a common spoken language. For example, American Sign Language (ASL) (Duarte et al., 2021; Shi et al., 2022; Uthus et al., 2023; Tanzer & Zhang, 2024; Li et al., 2020a) and Australian Sign Language (Auslan) (Shen et al., 2023; 2024a; Sheng et al., 2024) are linguistically distinct, each with unique phonological, lexical, and syntactic features. To bridge communication between Deaf and hearing communities, sign language translation (SLT) (Shen et al., 2025a) systems have been developed to automatically translate sign videos into spoken languages.

Among sign languages, fingerspelling (FS), the manual representation of alphabets and numbers, plays a critical role in SLT (Shen et al., 2023; Tanzer, 2024b; Georg et al., 2024; Kim et al., 2017; Papadimitriou et al., 2024; Sheng et al., 2026; Shen et al., 2024b), particularly for expressing proper nouns, technical terms, and items not represented in the standard sign lexicon. Unlike single-handed systems, such as ASL (Shi et al., 2021; Padden., 1998; Tanzer, 2024a), BANZSL[1] employs a distinc-

---

[1] **G BANZSL** refers to a sign language family which encompasses BSL, Auslan and NZSL. These sign languages can be considered as dialects of BANZSL due to their shared manual alphabet, grammatical structure, and substantial lexical overlap.

tive two-handed fingerspelling system. This two-handed system introduces significant challenges for machine translation (see the bottom row of Figure 2), such as frequent self-occlusion, high intra-letter variations, and rapid handshape transitions. Hence, accurate recognition of fingerspelling is crucial, as it frequently conveys essential semantic content, such as named entities, numerical data, and domain-specific vocabulary that lack conventional sign equivalents.

Despite significant progress in sign language research (Huang et al., 2018; Zhou et al., 2021; Shi et al., 2022; Duarte et al., 2021; Camgöz et al., 2018; Uthus et al., 2023; Tanzer & Zhang, 2024; Shen et al., 2023; 2024a), most publicly available datasets focus on single-handed fingerspelling understanding task, such as those used in ASL (Shi et al., 2021; Padden., 1998; Tanzer, 2024a; Georg et al., 2024) and GSL (Papadimitriou et al., 2024), leaving the two-handed fingerspelling system of BANZSL (Shen et al., 2024a; Prajwal et al., 2022) comparatively underexplored. Moreover, existing datasets often lack the scale and linguistic realism required for fingerspelling research. In particular, they rarely capture naturally occurring phenomena, such as spelling errors, lexical abbreviations, acronyms, and inline corrections, which are commonly encountered in practical scenarios. This highlights a critical gap: the need for a large-scale, real-world BANZSL fingerspelling dataset to facilitate the study on BSL, Auslan and NZSL.

To address this gap, we introduce **BANZ-FS**, a large-scale dataset dedicated to BANZSL fingerspelling, collected from both real-world and controlled environments. As shown in Figure 1, BANZ-FS integrates multiple sources to reflect diverse and authentic usage scenarios: (1) professional live Auslan interpretations from *ABC News with Auslan* broadcasts (capturing formal, high-register discourse); (2) controlled laboratory recordings (offering clean, high-quality reference data); and (3) user-generated vlog content from online platforms and social media (representing casual, daily communication). This diverse composition allows BANZ-FS to capture a broad spectrum of signing tempos and registers, from formal broadcast interpretation to everyday interaction.

Specifically, BANZ-FS comprises more than 35,000 aligned fingerspelling instances. During annotating fingerspelling, we additionally align 40 hours of Auslan news footage, which not only substantially extends the prior benchmark Auslan-Daily News (Shen et al., 2023) but also allows us to investigate recognition accuracy of fingerspelling within contexts. Our annotation protocol includes fine-grained alignment across video ↔ subtitles, video ↔ fingerspelled letters, and video ↔ target lexicons. As illustrated in Figure 2, we explicitly annotate and categorize key linguistic phenomena prevalent in fingerspelling, including abbreviations, acronyms, misspellings, and inline corrections. Furthermore, our proposed dataset captures the visual and articulatory complexities inherent in two-handed fingerspelling systems, underscoring the challenges of accurate fingerspelling recognition in BANZSL.

With BANZ-FS, we investigate a range of fingerspelling-related tasks, including fingerspelling detection, isolated fingerspelling recognition and fingerspelling recognition in context. We benchmark publicly available state-of-the-art models on each task and then report the performance using corresponding evaluation metrics. Experimental results demonstrate that the complexity and realism of BANZ-FS pose a significant challenge to existing methods, highlighting its potential to drive progress in two-handed fingerspelling understanding. Overall, the contributions of this work are threefold:

- We curate the first large-scale fingerspelling dataset specifically for the BANZSL system, capturing real-world complexities across diverse contexts.
- We provide comprehensive multi-level annotations to support fingerspelling-related tasks, including fingerspelling detection, isolated fingerspelling recognition, and fingerspelling recognition within continuous sign language sentences.
- We benchmark state-of-the-art methods to highlight the unique challenges posed by BANZ-FS, and establish an ideal platform to evaluate fingerspelling recognition capabilities.

## 2 RELATED WORK

### 2.1 FINGERSPELLING DATASETS

Early research in sign language recognition primarily addressed isolated sign language recognition (Shen et al., 2024a; Li et al., 2020a; Desai et al., 2023; Starner et al., 2023; Shen et al., 2025c;b), but recent trends have progressively emphasized continuous sign language recognition (Chen et al.,

Table 1: Comparison of the proposed **BANZ-FS** dataset with existing datasets widely used for fingerspelling-related tasks. "FSR-Context", "FSD ", and "IFSR" represent Fingerspelling Recognition in Context, Fingerspelling Detection and Isolated Fingerspelling Recognition, respectively.

| Dataset | SL | Video | Vocab. | # FS Seqs | #Signer | Source | FSR-Context | FSD | IFSR |
|---|---|---|---|---|---|---|---|---|---|
| Fleurs-ASL-FS (Tanzer, 2024a;b) | ASL | 1.7K | - | 0 | 5 | Lab | ✓ | | |
| SL-ReDu-Fing. (Papadimitriou et al., 2024) | GSL | 1.5K | 24 | 1.5K | 21 | Lab | | | ✓ |
| BOBSL-FS (Prajwal et al., 2022) | BSL | 5K | 26 | 5K | - | Web | | | ✓ |
| ChicagoFSVid (Kim et al., 2017) | ASL | 4K | 26 | 4K | 4 | Lab | | | ✓ |
| FSboard (Georg et al., 2024) | ASL | 151K | 36 | 151K | 147 | Smartphone | | | ✓ |
| ChicagoFSWild (Shi et al., 2018) | ASL | 7K | 26 | 7K | 160 | Web | | ✓ | ✓ |
| ChicagoFSWild+ (Shi et al., 2019) | ASL | 55K | 26 | 55K | 260 | Web | | ✓ | ✓ |
| Auslan-Daily Comm. (Shen et al., 2023) | Auslan | 14K | 3K | 1K | 49 | TV&Web | ✓ | ✓ | ✓ |
| Auslan-Daily News (Shen et al., 2023) | Auslan | 11K | 13K | 1K | 18 | TV&Web | ✓ | ✓ | ✓ |
| **BANZ-FS (Ours)** | **BANZSL** | **35K** | **36** | **35K** | **116** | **Lab&Web** | ✓ | ✓ | ✓ |

2022c; Min et al., 2021) and fingerspelling recognition (Shen et al., 2023; Kim et al., 2017; Shi et al., 2018; 2019; Fayyazsanavi et al., 2024). Despite growing interest, as shown in Table 1, existing fingerspelling datasets largely concentrate on American Sign Language (ASL) and single-handed signing systems (Papadimitriou et al., 2024), such as ChicagoFSWild+ (Shi et al., 2019), ChicagoFSWild (Shi et al., 2018), FSBoard (Georg et al., 2024) and Fleurs-ASL-FS (Tanzer, 2024a;b). Among these, FSBoard (Georg et al., 2024) is currently the largest dataset, containing approximately 151K fingerspelling sequences collected from 147 signers, captured uniquely via smartphone in a single-handed manner. However, FSBoard is limited to recognition tasks due to the absence of segment-level annotations, which restricts its application in fingerspelling detection. Datasets capturing fingerspelling "in-the-wild", such as ChicagoFSWild (Shi et al., 2018) and ChicagoFSWild+ (Shi et al., 2019), have improved realism by sourcing content from online platforms, encompassing diverse signer appearances and environmental variations. It is worth noting that Fleurs-ASL-FS (Tanzer, 2024a;b) only provides sentence-level annotations indicating the presence of fingerspelling, without corresponding temporal boundaries. As a result, it can only be used for fingerspelling in context task, but not for fingerspelling detection or localization.

Regarding BANZSL-related resources, prior datasets such as Auslan-Daily (Shen et al., 2023) and BOBSL-FS (Prajwal et al., 2022) contain only a limited number of fingerspelling instances, and consequently provide insufficient support for developing and evaluating fingerspelling-specific tasks. Our proposed **BANZ-FS** dataset addresses these limitations by introducing over 35K aligned fingerspelling instances with comprehensive annotations suitable for fingerspelling detection and recognition tasks. In parallel with the annotation of BANZ-FS, we also engaged Auslan experts to extend the Auslan-Daily (Shen et al., 2023) News subset through additional annotation and alignment, resulting in a threefold increase in scale.

## 2.2 FINGERSPELLING DETECTION AND RECOGNITION METHODS

Early fingerspelling detection (Shi et al., 2021) methods utilized visual features such as optical flow or predefined hand keypoints (Yang & Lee, 2010; Yanovich et al., 2016). However, such methods have primarily been evaluated in controlled environments, with limited effectiveness in unconstrained, real-world settings due to unreliable pose estimations (Tsechpenakis et al., 2006b;a). Recent approaches have favored recurrent neural networks (RNNs) (Luong et al., 2015) and transformer-based (Vaswani et al., 2017) architectures to enhance temporal modeling capabilities and robustness (Li et al., 2020c; Moryossef et al., 2020; Zuo et al., 2023; Pugeault & Bowden, 2011; Li et al., 2020a).

For fingerspelling recognition, convolutional neural networks (CNNs) combined with RNNs or Long Short-Term Memory (LSTM) networks have been widely employed (Schuster & Paliwal, 1997; Shi et al., 2021). Transformer-based models have recently emerged as powerful alternatives, effectively capturing long-range temporal dependencies and contextual information crucial for recognizing fingerspelled sequences (Boháček & Hrúz, 2022; Hu et al., 2024; Prajwal et al., 2022). Several studies have explored multimodal fusion approaches, integrating RGB frames, optical flow, and pose estimation features to enhance recognition accuracy (Jiang et al., 2021a;b; Zuo et al., 2023). While significant progress has been made, challenges remain, particularly regarding ambiguity due to similar handshapes across distinct letters and digits. Methods like those proposed in (Li et al., 2023;

Fayyazsanavi et al., 2024) modify Transformer encoder-decoder architectures explicitly to mitigate ambiguities arising from visually similar fingerspelling representations.

# 3 BANZ-FS DATASET

In this section, we describe data collection for web-based fingerspelling data, as well as the recording protocol for the lab-collected instances[2]. We provide detailed statistics of the **BANZ-FS** dataset.

## 3.1 COLLECTION, CLEANING AND LABELLING PROCEDURE FOR WEB DATA

**Collection.** "ABC News with Auslan" and YouTube sources are open sources. Beginning in 2022, "ABC News with Auslan" has provided weekly broadcasts covering key domestic and international news events, as well as weather forecasts. It is an ongoing program, and previous work (Shen et al., 2023) aligned 45 videos collected up to May 2023, along with a small number of fingerspelling annotations. In this work, we extend the collection by acquiring an additional 80 videos spanning from May 2023 to April 2025. These broadcasts feature live sign language translation (simultaneous interpretation) by Auslan experts, intended for deaf and hard-of-hearing viewers. News content inherently includes a rich set of fingerspelling scenarios, such as personal names, place names, organization names, phone numbers, and other proper nouns, making it an ideal source for studying fingerspelling phenomena. To further diversify our dataset, we also include several high-quality publicly available documentaries and educational videos interpreted with BSL and NZSL, primarily sourced from YouTube. These videos typically feature daily conversations, learning activities, or introductions to specific topics. All online videos are included only via their official public URLs, without redistribution of any copyrighted material, as detailed in the Ethics Statement.

**Cleaning.** All original videos are accompanied by standard English dubbing and subtitles. We retrieve subtitles for each complete video, formatted as "*[Start Time] subtitle [End Time]*", with timing aligned to the spoken dubbing. Following the subtitle cleaning strategy proposed in Auslan-Daily (Shen et al., 2023), we perform simple cleaning operations: merge fragments ending in commas with their subsequent lines, split overlapping subtitles into separate sentences, and discard entries with only non-semantic fillers. As a result, we obtain approximately 30K complete and cleaned subtitles requiring further alignment with fingerspelling segments.

**Labelling.** To annotate fingerspelling instances, we invited experienced Auslan experts to assist in the labelling process. In particular, for news videos, we additionally perform temporal alignment to ensure segment-level consistency. For each video, we first employ AlphaPose (Fang et al., 2022; 2017; Li et al., 2019) to track all individuals in the scene. The annotation process proceeds in several steps: (1) verify and refine video-subtitle alignment; (2) identify the signer ID based on pose trajectories. AlphaPose tracks all people in the scene and generates identity-consistent pose sequences. Annotators manually review these trajectories and select the signer based on spatial position and continuous signing motion; (3) if fingerspelling is present, annotate the corresponding temporal segment; and (4) retrieve the associated target lexicon from the subtitle, if it exists. To guarantee the annotation quality of our dataset, we conduct a cross-check verification process during each data labelling procedure stage. Specifically, we employed a "recognition-based verification" protocol where each Auslan annotator (examiner) cross-checks a random 5% sample of clips provided by another annotator. In practice, we observed high consistency, with approximately 95% of the sampled batches passing verification in the first round. If the examiner finds more than 10% of annotated videos have obvious errors, the entire batch is rejected, and a third annotator is invited to review and correct the annotations. Through the collaborative efforts of five Auslan experts and five annotators, we complete all annotations with approximately 500 work hours. Overall, our dataset contains the following **annotations**: (1) temporal boundaries of sign video clips; (2) temporal boundaries of fingerspellings; (3) lexical forms of fingerspellings; and (4) English transcriptions. These annotations can be further investigated for fingerspelling-related tasks.

---

[2]The original Auslan News data is provided by Auslan-Daily (Shen et al., 2023). Both the web-based and lab-collected portions of our dataset will be released under the **Creative Commons BY-NC-SA 4.0** license ©.

Table 2: Key statistics of the BANZ-FS dataset across three data sources: ABC News with Auslan, Lab Capture, and YouTube. OOS (out-of-training Signers) and OOFS (out-of-training FS strings) are signers and FS sequences that never appear in the training set, while FS Singletons occur only once in training.

| Data Source | ABC News with Auslan | | | Lab Capture | | | YouTube | | | |
|---|---|---|---|---|---|---|---|---|---|---|
| Source Language | Auslan | | | - | | | BSL&NZSL | | | |
| Domain/Topic | News | | | Daily Used | | | Communication | | | |
| Video Resolution@FPS | 1280×720@25 | | | 1920×1080@30/1280×720@60 | | | Various | | | |
| Split | Train | Dev | Test | Train | Dev | Test | Train | Dev | Test | Total |
| Video Segments | 18,694 | 1,608 | 1,896 | 6,828 | 1,952 | 1,952 | 1,498 | 300 | 300 | 35,028 |
| Signers | 24 | 22 | 19 | 65 | 52 | 50 | 18 | 10 | 7 | 116 |
| Tot. OOSs | - | 5 | 5 | - | 10 | 12 | - | 3 | 2 | 21 |
| Avg. FS Segments | 2.4 | 2.0 | 2.5 | 1.0 | 1.0 | 1.0 | 2.3 | 1.8 | 2.0 | 1.95 |
| Tot. FS Chars | 144,090 | 12,383 | 15,717 | 11,748 | 3,360 | 3,360 | 12,034 | 2,154 | 2,446 | 207,292 |
| Avg. FS Speed (chars/sencond) | 4.59 | 4.89 | 4.17 | 1.30 | 1.31 | 1.30 | 1.99 | 1.45 | 1.68 | 3.41 |
| Tot. OOFSs | - | 304 | 450 | - | 0 | 0 | - | 43 | 64 | 813 |
| FS Singletons | 1,201 | - | - | 0 | - | - | 0 | - | - | 1,191 |

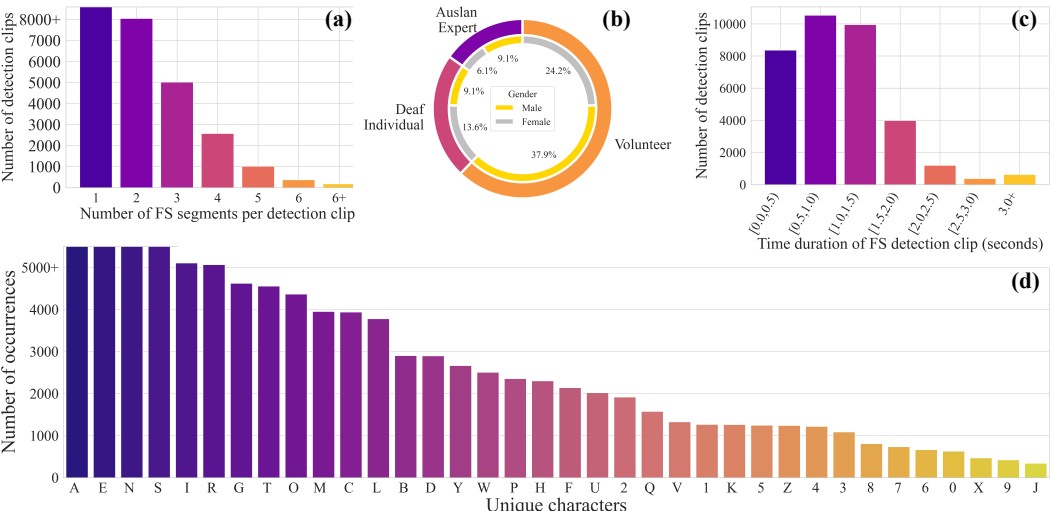

Figure 3: (a) Distribution of the number of fingerspelling (FS) segments per clip. (b) Distribution of FS clip durations. (c) Distribution of signer demographics categorized by Auslan proficiency[3]and gender. (d) Character frequency distribution across all FS clips.

## 3.2 COLLECTION FOR LAB DATA

To complement our web data, followed by (Shen et al., 2024a; Ying et al., 2024), we record lab-controlled videos using a multi-camera RGB-D setup. The recording studio is equipped with a green screen and includes three Kinect-V2 cameras positioned at left-front, front, and right-front angles, along with a centrally placed RealSense camera. We invite participants with diverse Auslan experience, including deaf individuals, Auslan experts, and sign language learners. Participants are instructed to perform frequently used fingerspelling words and expressions commonly encountered in daily communication contexts. Each sign instance is verified by at least one expert to ensure expression accuracy, while the inclusion of volunteers enhances signer diversity and realism. This setup facilitates the study of cross-camera robustness and supports high-quality benchmarking under controlled conditions. Full details regarding ethical compliance, participant consent, and fair compensation are provided in the Section Ethics Statement.

---

[3]In this dataset, the label "Auslan Expert" refers specifically to hearing professional interpreters, whereas "Deaf individual" denotes Deaf participants. These labels reflect only the demographic grouping used in this study and do not imply that the categories are mutually exclusive.

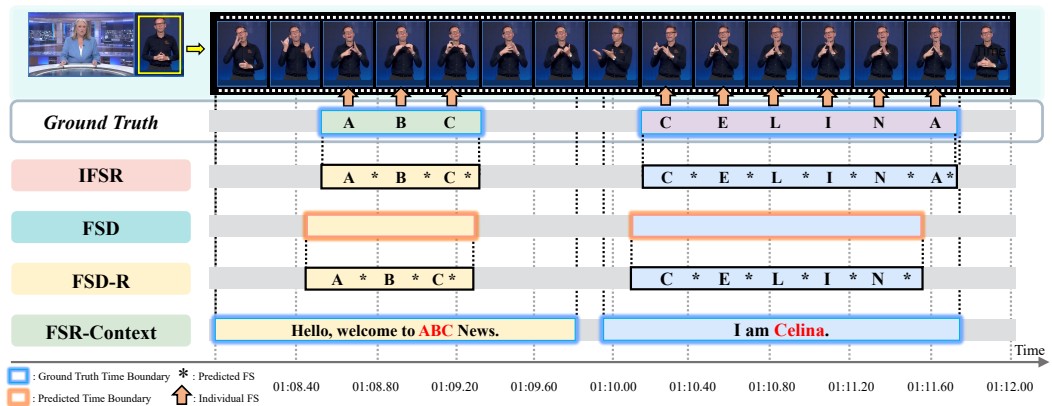

Figure 4: Overview of fingerspelling-related tasks in our BANZ-FS dataset.

## 3.3 DATA STATISTICS

As shown in Table 2, we present key statistics of BANZ-FS to highlight the diversity and complexity of the dataset. BANZ-FS consists of over 35,000 annotated video segments sourced from news broadcasts, lab recordings, and online videos, covering 116 unique signers. The dataset is split into training, development, and test sets to facilitate fair evaluation of fingerspelling-related tasks. We segment each video by applying a 10-second sliding window around any detected FS segment. As a result, each detection clip may contain multiple FS instances. As shown in Figure 3, most clips contain only 1–2 FS segments and last less than 1.5 seconds, indicating that FS is often embedded briefly within continuous signing. Furthermore, the signer population includes a balanced mix of Auslan experts, deaf individuals, and volunteers, offering a wide range of signing styles and linguistic competence. The FS character distribution reveals a long-tail pattern: common letters such as A, E, and N appear frequently, while rare characters (e.g., numerals and less frequent letters) occur sparsely. This imbalance poses additional challenges for generalization and open-vocabulary recognition, especially in low-resource conditions. In addition, we report the number of out-of-training FS strings (OOFS) and FS singletons in Table 2, which quantify the presence of unseen or rare FS sequences and further reflect the open-set nature of the task. To evaluate generalization to unseen users, we explicitly report the number of out-of-training signers in Table 2 which refers to signers that never appear in the training set. Further statistics and discussions can be found in Appendix Section B, Section C.6, Section C.7 and Section H.1.

## 4 OVERVIEW OF BANZ-FS TASKS AND EVALUATION METRICS

In this section, we provide an overview of the **BANZ-FS** benchmark tasks and their corresponding evaluation metrics, as illustrated in Figure 4.

**Isolated Fingerspelling Recognition (IFSR)** (Shi et al., 2018; 2019): Given a segmented fingerspelling clip $\mathbb{V}_{fs} = \{I_{f_s}, ..., I_{f_e}\}$, the goal of IFSR is to transcribe it into the corresponding letter sequence $\hat{L} = \{l_1, ..., l_n\}$. Evaluation Metric for IFSR is **Letter Accuracy** - Defined as $1 - \frac{\text{EditDistance}(L^*, \hat{L})}{|L^*|}$, where $L^*$ is the ground-truth letter sequence and $\hat{L}$ is the predicted sequence. This edit-distance-based metric captures the correctness of the full predicted sequence, accounting for insertions, deletions, and substitutions.

**Fingerspelling Detection (FSD)** (Shi et al., 2021): Given an untrimmed sign language video $\mathbb{V} = \{I_1, I_2, ..., I_T\}$ with $T$ frames, the goal of FSD is to identify temporal segments $(f_s, f_e)$ that localize fingerspelling intervals within $\mathbb{V}$. Each predicted segment corresponds to a time span where fingerspelling occurs. The evaluation metric for FSD is **AP@IoU**, specifically **AP@IoU**$_{0.5}$. It represents the Average Precision calculated based on a temporal Intersection-over-Union (IoU) threshold of 0.5 between predicted and ground-truth segments.

**Fingerspelling Detection followed by Recognition (FSD-R)** (Shi et al., 2021): FSD-R is a two-stage approach where an FSD model first predicts temporal segments from an untrimmed sign language video $\mathbb{V}$, and each predicted segment is subsequently processed by a fingerspelling recognizer (IFSR) to generate the corresponding letter sequence. The evaluation metric for FSD-R is **AP@Acc**, specifically **AP@Acc$_{0.5}$**. It represents the Average Precision where a predicted segment is considered a True Positive only if the character-level accuracy of the downstream recognizer exceeds a threshold of 50%.

**Fingerspelling Recognition in Context (FSR-Context)** (Tanzer, 2024b): Given a full sentence-level sign language video $\mathbb{V}$ and its predicted spoken language translation $\hat{T}$, the task of FSR-Context is to evaluate how accurately the model transcribes fingerspelled terms embedded within the sentence. Specifically, fingerspelled spans annotated in the video are aligned with corresponding spans in the predicted translation, and character-level accuracy is measured. Evaluation Metrics for FSR-Context is Letter Accuracy.

## 5 BANZ-FS Benchmark

### 5.1 Video Representation

**Pose-based video feature representation:** Pose-based representations are robust against background clutters, lighting conditions, and occlusions, while explicitly depicting human hand and limb movements (Weinzaepfel et al., 2015; Si et al., 2018; Yan et al., 2018). Several recent studies exploit pose information and achieve state-of-the-art performance in fingerspelling-related tasks (Tanzer, 2024b; Fayyazsanavi et al., 2024; Moryossef et al., 2023). Hence, we use the key points extracted from DWPose (Yang et al., 2023) as video features to provide benchmark results.

**RGB-based video feature representation:** Several models directly extract features from sign videos, such as CNN-RNN-HMM network (Camgöz et al., 2018), S3D (Chen et al., 2022b) and I3D (Carreira & Zisserman, 2017). In the works (Albanie et al., 2020; Li et al., 2020a;b), I3D is used for sign video representation. To better adapt to SL dataset and capture the spatio-temporal information of signs, inspired by (Li et al., 2020b), we finetune I3D on a word-level sign language recognition dataset (Shen et al., 2024a) and extract sign video features with different window widths and strides. Recent studies have shown that feeding raw RGB video directly into end-to-end models yields strong performance on various sign language tasks (Chen et al., 2022c; Min et al., 2021). Following this trend, we adopt raw video frames as input for our end-to-end models.

### 5.2 Benchmark Results

In this section, we provide benchmark results of isolated fingerspelling recognition (IFSR), fingerspelling detection (FSD), fingerspelling detection followed by recognition (FSD-R) and fingerspelling recognition in context (FSR-Context) tasks on BANZ-FS.

**Isolated Fingerspelling Recognition (IFSR):** Table 3 presents a cross-domain evaluation of state-of-the-art models on the IFSR task. The models are trained separately on four different subsets—News, Lab, Web, and the union of all (Full)—and evaluated across each domain. This setup allows for a detailed analysis of both in-domain performance (training and testing on the same source) and out-of-domain generalization (training on one domain, testing on another). Among all the models, HandReader Korotaev et al. (2025) consistently demonstrates the strongest cross-domain robustness. When trained on the Full set, it achieves the highest overall accuracy (75.4%), and outperforms other models by a large margin on challenging domains such as Web (40.2%). Notably, HandReader (Korotaev et al., 2025) trained on News alone generalizes well to Lab data (48.5%) and achieves the best News-to-News performance (68.3%), reflecting the benefit of its multimodal (RGB + 3D pose) architecture in capturing signer-invariant features. In contrast, Iterative-Att (Shi et al., 2019) show limited generalization ability. The performance drops significantly when evaluated on unseen domains, especially on Web videos where visual variability is high. MiCT-RANet (Mahoudeau, 2020) and TS-FS-Reg (Chen et al., 2022c) achieve stronger generalization than early models, particularly when trained on the Full set, with accuracies of 68.6% and 69.7% respectively. TS-FS-Reg (Chen et al., 2022c) benefits from dual-modality inputs, which help mitigate overfitting to domain-specific appearance. Overall, the results reveal that while most models perform well on the domain they

Table 3: Performance comparison (Letter Accuracy) on Fingerspelling Recognition (IFSR) across three data domains: News, Lab, and Web. "Full" refer to the combined dataset.

| Method | Train | Letter Accuracy (%) | | | |
|---|---|---|---|---|---|
| | | News | Lab | Web | Full |
| Iterative-Att (Shi et al., 2019) | Full | 45.6 | 72.3 | 51.3 | 58.6 |
| | News | 50.6 | 44.2 | 38.4 | 46.7 |
| | Lab | 30.2 | 87.7 | 30.3 | 57.3 |
| | Web | 21.0 | 35.8 | 36.3 | 29.1 |
| MiCT-RANet (Mahoudeau, 2020) | Full | 56.4 | 81.8 | 60.1 | 68.6 |
| | News | 57.2 | 51.0 | 44.8 | 53.3 |
| | Lab | 33.2 | 92.3 | 31.8 | 60.9 |
| | Web | 22.9 | 38.6 | 42.5 | 31.7 |
| TS-FS-Reg (Chen et al., 2022c) | Full | 57.2 | 82.9 | 62.4 | 69.7 |
| | News | 59.2 | 54.3 | 50.4 | 56.2 |
| | Lab | 39.7 | 92.3 | 34.8 | 64.1 |
| | Web | 24.1 | 36.8 | 46.2 | 31.6 |
| FS-PoseNet (Fayyazsanavi et al., 2024) | Full | 62.5 | 87.3 | 70.1 | 74.7 |
| | News | 66.2 | 52.9 | 48.2 | 58.6 |
| | Lab | 36.2 | 92.8 | 29.4 | 62.3 |
| | Web | 26.5 | 47.2 | 51.4 | 38.0 |
| HandReader (Korotaev et al., 2025) | Full | 64.4 | 86.7 | **71.8** | **75.4** |
| | News | **68.3** | 55.6 | 48.5 | 60.8 |
| | Lab | 37.1 | **93.1** | 32.6 | 63.1 |
| | Web | 29.8 | 48.1 | 55.0 | 40.2 |

Table 4: Performance comparison on Fingerspelling Detection (FSD) and FSD-R tasks across three data domains. Average Precision (AP) at 0.5 IoU threshold for FSD ($AP@IoU_{0.5}$), and AP at 0.5 recognition accuracy threshold for FSD-R ($AP@Acc_{0.5}$).

| Method | Train | $AP@IoU_{0.5}$ | | | | $AP@Acc_{0.5}$ | | | |
|---|---|---|---|---|---|---|---|---|---|
| | | News | Lab | Web | Full | News | Lab | Web | Full |
| Bi-LSTM CTC (Huang et al., 2015) | Full | 31.1 | 56.0 | 27.2 | 42.5 | 15.5 | 39.9 | 14.8 | 26.9 |
| | News | 43.4 | 29.0 | 23.8 | 35.2 | 21.6 | 15.6 | 8.5 | 17.8 |
| | Lab | 10.0 | 72.9 | 15.1 | 40.0 | 5.0 | 71.1 | 7.5 | 36.3 |
| | Web | 13.1 | 37.4 | 26.1 | 25.5 | 6.5 | 13.2 | 13.3 | 10.1 |
| Modified R-C3D (Xu et al., 2017) | Full | 35.2 | 64.6 | 32.2 | 48.8 | 19.6 | 43.3 | 15.6 | 30.5 |
| | News | 47.9 | 33.0 | 25.0 | 39.2 | 23.2 | 16.2 | 9.3 | 18.9 |
| | Lab | 11.9 | 75.5 | 17.0 | 42.2 | 6.1 | 77.3 | 10.8 | 39.9 |
| | Web | 15.9 | 40.9 | 30.1 | 28.7 | 7.0 | 14.5 | 14.4 | 11.1 |
| TS-FS-Det (Chen et al., 2022c) | Full | 41.3 | 69.0 | 37.4 | 54.1 | 23.5 | 64.0 | 22.1 | 42.5 |
| | News | 53.6 | 41.8 | 34.0 | 46.6 | 29.3 | 26.5 | 19.3 | 27.3 |
| | Lab | 12.2 | 79.6 | 18.7 | 44.4 | 7.4 | 77.9 | 9.0 | 40.7 |
| | Web | 19.6 | 44.7 | 37.5 | 32.7 | 9.9 | 23.9 | 22.2 | 17.4 |
| MT-FS-Det (Shi et al., 2021) | Full | 48.6 | 79.7 | 41.6 | 62.7 | 25.4 | 68.7 | 26.8 | 45.9 |
| | News | 57.8 | 44.2 | 34.2 | 49.7 | 32.9 | 29.1 | 20.9 | 30.2 |
| | Lab | 18.1 | 82.3 | 19.0 | 48.4 | 10.4 | 81.7 | 13.7 | 44.2 |
| | Web | 20.8 | 44.2 | 41.5 | 33.3 | 10.1 | 25.5 | 26.9 | 18.6 |
| SL-Seg (Moryossef et al., 2023) | Full | 53.9 | 82.7 | **47.3** | **66.9** | 33.7 | 76.3 | **30.2** | **53.5** |
| | News | **60.0** | 45.2 | 38.8 | 51.5 | **35.9** | 30.2 | 24.2 | 32.3 |
| | Lab | 18.8 | **86.3** | 21.5 | 50.7 | 10.2 | **85.0** | 14.9 | 45.7 |
| | Web | 22.5 | 46.1 | 40.4 | 34.9 | 11.8 | 28.2 | 29.9 | 20.8 |

are trained on—especially Lab, which offers controlled recording conditions—cross-domain generalization remains a significant challenge. FS-PoseNet (Fayyazsanavi et al., 2024) stands out by consistently maintaining strong performance across all domains, making it particularly promising for deployment in real-world scenarios with diverse video sources.

**Fingerspelling Detection (FSD):** The FSD results in Table 4 show notable variation across domains. SL-Seg (Moryossef et al., 2023) achieves the best overall detection performance, particularly excelling on the Web domain (47.3%), where other models generally struggle. This suggests that frame-level BIO tagging with pose-based cues provides more robust temporal boundary modeling than proposal-based or regression-based methods. While MT-FS-Det (Shi et al., 2021) and TS-FS-Det (Chen et al., 2022c) perform competitively on News and Lab subsets, their generalization to noisy Web data is limited. In contrast, performance of earlier approaches, such as Modified R-C3D (Xu et al., 2017) and Bi-LSTM CTC (Huang et al., 2015), is much inferior, highlighting the importance of structured temporal representations and boundary-aware learning for reliable fingerspelling localization.

**Fingerspelling Detection followed by Recognition (FSD-R):** The FSD-R results evaluate the quality of detected segments by measuring whether they are both correctly localized and correctly recognized. In our setup, each predicted segment is fed into a pre-trained FS-PoseNet (Fayyazsanavi et al., 2024) recognizer, and is counted as correct only if the recognition accuracy exceeds 50%. As shown in Table 4, we observe a clear performance gap between detection and FSD-R. Even well-localized segments often fail to reach the required recognition threshold, especially in the Web domain. SL-Seg (Moryossef et al., 2023) achieves the highest overall AP@Acc (53.5%), yet still struggles on the Web subset (20.8%), underscoring the difficulty of maintaining segment quality under noisy conditions. These results emphasize that effective detection must also consider recognizability, motivating future work on recognition-aware detection or joint optimization approaches.

**Fingerspelling Recognition in Context (FSR-Context):** Following the protocol introduced in (Tanzer, 2024b), we extract fingerspelled spans from predicted translations and compute character-level Letter Accuracy based on alignment with annotated ground-truth phrases. We conduct our evaluation on expanded Auslan News dataset, which contains 18,604 sentence-level sign language videos, 13,208 of which include fingerspelled terms. We first evaluate a state-of-the-art gloss-free SLT model (Zhou et al., 2023) on sentence-level sign language videos that contain fingerspelled content. The model achieves a Letter Accuracy of 16.4% on the FSR-Context task. We then compare two Transformer-based translation models: T5 (Raffel et al., 2020) (subword tokenization) and ByT5 (Xue et al., 2022) (character-level tokenization). Results show that ByT5 outperforms T5,

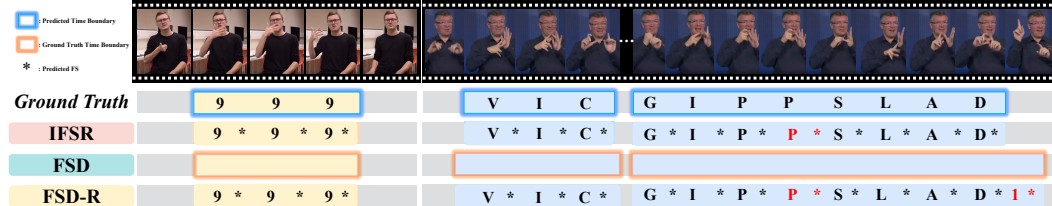

Figure 5: Case study comparing IFSR, FSD, and FSD-R on fingerspelling sequences.

achieving a Letter Accuracy of 25.8% compared to just 10.2% for T5. These findings are consistent with previous work (Tanzer, 2024b) and suggest that character-level tokenization offers substantial benefits for preserving the spelling of out-of-vocabulary words in translation output. The details of the expanded Auslan News dataset are provided in the Appendix Section C.3.

## 5.3 CASE STUDY

In Figure 5, we present several case studies across IFSR, FSD, and FSD-R settings. Some segments are correctly detected and recognized. However, rapid fingerspelling challenges the recognizer (*e.g.*, two P's within 8 frames), and loose detection boundaries cause unrelated gestures to be included, resulting in spurious predictions like "**1**". These errors highlight the compound difficulty of accurate detection and recognition under natural signing conditions. Additional case studies are provided in the Appendix Section I.

## 6 CONCLUSION

In this work, we introduce **BANZ-FS**, a large-scale and richly annotated dataset dedicated to fingerspelling in the BANZSL. Our dataset is constructed from three diverse sources that cover a broad spectrum of signing tempos and registers, from formal broadcast interpretation to everyday interaction. BANZ-FS features over 35,000 aligned fingerspelling instances with multi-level annotations, including temporal segments, character sequences, lexical forms, and full-sentence transcriptions. We benchmark several fingerspelling-related tasks on BANZ-FS, including fingerspelling detection, isolated fingerspelling recognition, and fingerspelling recognition in context. To support contextual FS evaluation, we also extend the Auslan-Daily News subset with three times more aligned content. Through analysis and experiments, we demonstrate the challenges and opportunities posed by fingerspelling in BANZSL, particularly in the context of two-handed systems, self-occlusion, rapid transitions, and lexical variability. We hope that BANZ-FS will serve as a valuable resource for advancing sign language understanding, and encourage further research on fingerspelling phenomena across diverse linguistic and visual contexts.

## ACKNOWLEDGEMENT

This research is funded in part by ARC-Discovery grant (DP220100800 to XY), ARC-DECRA grant (DE230100477 to XY) and Google Research Scholar Program. We also gratefully thank all the anonymous reviewers and ACs for their constructive comments.

## ETHICS STATEMENT

This work involves the curation of BANZ-FS, a large-scale dataset for BANZSL fingerspelling. We have carefully considered the ethical implications of data collection, annotation, and release, in line with the ICLR Code of Ethics.

**Human Subjects & Consent.** All data collected in the *Lab Recordings* subset are conducted under ethical oversight in a safe, supervised laboratory environment. Each participant (or guardian, in the case of minors) signs a detailed consent form (Appendix Section D) stating that their facial expressions and hand gestures may be recorded for research use only, without commercial redistribution.

Participation is voluntary, and participants can withdraw at any time. The study protocol, consent procedure, and data handling plan are reviewed by the *University's Research Ethics Committee*, which classifies the study as ethically exempt under its guidelines. We maintain signed consent records and perform post-recording verification, ensuring adherence to consent terms.

**Compensation.** All contributors are fairly compensated: general volunteers at AUD $40/hour, and Deaf signers and Auslan experts at AUD $100/hour, in line with institutional and regional standards. Approximately 500 hours of paid annotation work are carried out under formal contract.

**Privacy & Anonymization.** All participants explicitly consent to the public release of their (un-blurred) recordings for academic use. A withdrawal and anonymization protocol is in place: we apply face-blurring (using `deface`) upon participant request, and delete data entirely if consent is withdrawn. Future releases also provide 2D/3D pose annotations to support privacy-preserving research. No personally identifiable information (PII) or sensitive data (health or financial information) is collected.

**Copyright & Licensing.** For the *web-sourced* and *broadcast* portions of the dataset, we strictly comply with platform Terms of Service. We do not download, re-host, redistribute, or store any copyrighted audiovisual content from YouTube or other platforms. Only our derived annotations and the official public URLs of the original videos are released, and all video playback remains hosted under the control of the original content creator. Users of BANZ-FS are required to adhere to the corresponding platform ToS. Where videos are distributed under Creative Commons licenses (e.g., CC BY-NC-SA 4.0) (Shen et al., 2023) or explicit permissions are available, we follow those terms accordingly. The full dataset release is governed by a CC BY-NC-SA 4.0 license and an End-User License Agreement (EULA) that prohibits commercial exploitation, re-identification, or surveillance use. A takedown-request email is provided for removal or anonymization requests.

**Fairness & Representativeness.** We report signer demographics (gender, age, region, race) across all three data sources (Appendix Section C.7). While Auslan data dominates, the BANZSL alphabet is shared across dialects, and cross-dialect evaluation shows good generalization. We acknowledge remaining imbalances and plan to mine additional BSL/NZSL data and incorporate community feedback in future releases.

**Responsible Use.** We include a Responsible Use Statement with the dataset that explicitly prohibits its deployment in surveillance, biometric identification, or other sensitive decision-making contexts without further ethical review. Our release aims to support inclusive, equitable research benefiting the BANZSL and Deaf communities.

## REPRODUCIBILITY STATEMENT

We take reproducibility seriously, and due to the current stage of the review process, we provide a website that includes some data samples (due to storage limitations), data storage structure, and annotation files. Our main dataset repository and Google Drive link will be released after the review process. This ensures that anyone can regenerate exactly the same dataset version used in our experiments.

For benchmarking, we exclusively use publicly available models and follow their original hyperparameter settings without any modification. This guarantees fair and consistent comparison across methods. Detailed dataset statistics, preprocessing steps, and quality control procedures are provided in Section 3. For more experiment details, please refer to Appendix Section F and our website ⍟ BANZ-FS.

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

This appendix is organized as follows:

- BROADER IMPACT (Section A).
- LIMITATION AND FUTURE WORK (Section B).
- BUILDING BANZ-FS (Section C).
- CONSENT FORM FOR BANZ-FS RECORDING (Section D).
- MORE DETAILS FOR VIDEO REPRESENTATION (Section E).
- EXPERIMENTAL SETTINGS (Section F).
- THE BASELINE OF AUSLAN-DAILY NEWS V2 (Section G).
- ADDITIONAL DISCUSSION (Section H).
- CASE STUDY FOR BANZ-FS FINGERSPELLING DETECTION AND RECOGNITION (Section I).
- CASE STUDY FOR AUSLAN-DAILY NEWS SIGN LANGUAGE TRANSLATION (Section J).
- LLM USAGE STATEMENT (Section K).

## A  BROADER IMPACT

The BANZ-FS dataset addresses a critical gap in the sign language research community by focusing on the underrepresented two-handed fingerspelling systems of BANZSL (British, Australian, and New Zealand Sign Languages). This work has the potential to significantly improve accessibility technologies for Deaf and hard-of-hearing communities across multiple English-speaking regions. By supporting robust research on fingerspelling detection and recognition, BANZ-FS can contribute to the development of real-time translation systems, assistive educational tools for sign language learners, and inclusive communication platforms. Importantly, the dataset includes real-world scenarios drawn from news, vlogs, and lab settings, thus promoting domain generalization in practical applications. However, as with any dataset involving human subjects, privacy, representation, and consent are essential considerations. All included video data are sourced from publicly available content or recorded with informed consent. Nevertheless, we acknowledge that biases may still exist, such as overrepresentation of Auslan compared to BSL or NZSL, and uneven distribution of fingerspelled letters. Future work should address these issues through targeted data collection and model adaptation techniques. Finally, while the goal is to aid accessibility, there is also a risk of misuse—such as surveillance or unauthorized profiling using sign language recognition systems. We strongly encourage researchers and practitioners to follow ethical guidelines and collaborate closely with Deaf communities when deploying models trained on BANZ-FS.

## B  LIMITATION AND FUTURE WORK

Our work has two primary limitations. **Letter Frequency Imbalance.** Letter imbalance exists in the dataset—frequent characters like "N", "S", "E", and "W" dominate due to their linguistic role in directional terms, while rarer letters such as "X" and "J" are significantly underrepresented. Such skew is common in natural language corpora and other fingerspelling datasets; for example, in ChicagoFSWild+ (Shi et al., 2019), the most frequent letters "A", "E", and "O" each appear over 20K times, whereas the rarest letters "Z" and "Q" appear only 494 and 311 times, respectively. To mitigate this effect, we carefully balanced the validation and test splits to ensure fair representation of low-frequency letters. As shown in Table 5, several characters and digits occur fewer than 1,000 times, but we ensure their presence across all splits. Our dataset design further supports alleviation of long-tail issues: the Lab Recordings subset enables targeted collection of underrepresented characters, and our BANZ-FS-trained detector can be used to mine additional candidate instances from unlabeled videos. In future work, we plan to explore sign language generation techniques to synthetically augment low-frequency letters and improve overall letter coverage. **Dialect Imbalance.** There is a clear dialect imbalance within the BANZSL subset: the data is heavily skewed toward Auslan, with BSL and NZSL contributions being relatively limited. Nevertheless, all three dialects share the same two-handed fingerspelling alphabet, and as shown in Table 3 and Table 4, models trained on the

Table 5: Frequency of underrepresented letters and digits (total count $< 1,000$) across train/valid/test splits in BANZ-FS.

| Split | FS-"8" | FS-"7" | FS-"6" | FS-"0" | FS-"X" | FS-"9" | FS-"J" |
|---|---|---|---|---|---|---|---|
| Train | 596 | 538 | 503 | 464 | 348 | 324 | 271 |
| Valid | 98 | 89 | 76 | 84 | 56 | 42 | 29 |
| Test | 114 | 107 | 85 | 79 | 66 | 54 | 39 |
| **Total** | 808 | 734 | 664 | 627 | 470 | 420 | 339 |

Auslan subset generalize well to BSL and NZSL web data. In future work, we will use our trained BANZ-FS detector to mine additional BSL/NZSL clips from broadcast and web sources, thereby improving dialectal diversity and representation.

Beyond data, we also plan to investigate recognition-aware detection models that jointly optimize for temporal localization and fingerspelling accuracy. Our current benchmark focuses on RGB front-view videos to ensure comparability with web data and existing open-source models. Although the lab setup records synchronized multi-view RGB-D streams, these modalities are not used in the reported experiments. In future work, we plan to explore depth-based modeling, multi-view fusion, and 3D hand reconstruction to further leverage the full potential of the released lab recordings. In addition, we aim to explore context-conditioned models that leverage sentence-level semantics to improve recognition of ambiguous or incomplete fingerspelling segments.

## C    BUILDING BANZ-FS

In this section, we explain the various stages of data processing and labelling in detail for preparing BANZ-FS, from collecting sources to storing final data.

### C.1    DATA PROCESSING AND LABELLING

Although the original Auslan videos are accompanied by English subtitles, the sentence boundaries in the subtitles are often misaligned with the actual signing segments due to differences in grammar, timing, and expression modalities between sign and spoken languages. To address this, we perform a sentence-level alignment procedure. First, we clean the raw subtitles by merging incomplete fragments (*e.g.*, those ending with commas), splitting multiple complete sentences within a single time interval, and removing non-informative expressions such as interjections.

As delineated in Section 3.1, we employ three distinct operations for subtitle cleaning (Shen et al., 2023). Here, we present a few representative examples:

- Incomplete subtitles:
  *[00:00:07,480]-[00:00:10,160] Today, a flood emergency warning issued for*
  *[00:00:10,200]-[00:00:18,720] Tasmania's River Derwent.*
  Revise:
  *[00:00:07,480]-[00:00:18,720] Today, a flood emergency warning issued for Tasmania's River Derwent.*

- Several complete subtitles that appear within a time interval:
  *[00:00:54,520]-[00:00:59,320] Hello and welcome to ABC News. I'm Gemma Veness.*
  Revise:
  *[00:00:54,520]-[00:00:59,320] Hello and welcome to ABC News.*
  *[00:00:54,520]-[00:00:59,320] I'm Gemma Veness.*

- Complete sentence that only contains modal particles:
  *[00:23:42,240]-[00:23:43,080] Ha-Ha.*
  Revise:
  Remove this subtitle.

After preprocessing, we obtain clean sentences. Then, Auslan experts manually align each sentence with its corresponding video segment. As illustrated in Figure 6, this alignment is conducted at the sentence level, taking into account both audio-aligned and sign-aligned timelines. The signer is first tracked, followed by precise segmentation of each sentence's temporal span within the video.

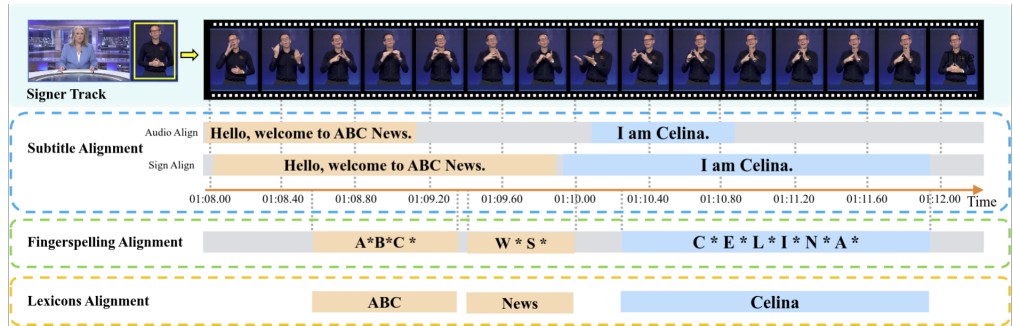

Figure 6: Illustration of sentence-level alignment. The signer is first tracked, followed by audio and sign alignment of the subtitle. Sentence segments, fingerspelling intervals, and lexical boundaries are then precisely annotated.

This results in temporally grounded video–sentence pairs, which are essential for training robust and accurate sign language translation (SLT) models.

In addition to sentence-level alignment, we also annotate fine-grained elements such as fingerspelling segments and lexical boundaries. Fingerspelling alignment identifies the exact start and end times of letter-by-letter signing, which often corresponds to proper nouns or unseen words. Lexicon alignment further breaks down the sentence into semantically significant units such as named entities or domain-specific terms. These multi-level annotations enable a richer understanding of signed content and facilitate downstream tasks such as fingerspelling recognition and detection.

## C.2    ADDITIONAL DETAILS ON DATASET STRATIFICATION

This section provides additional clarification on how the dataset splits are constructed.

**Signer stratification.** Each domain includes Out-of-Set (OOS) signers in the test partition to prevent identity memorization. News, Lab, and Web respectively contain multiple unseen signers in their test splits, ensuring that evaluations reflect generalization to new users rather than repeated individuals.

**Vocabulary stratification (OOFS).** We track Out-of-Training Fingerspelling Strings (OOFS), defined as character sequences appearing in the test set but not in the training set. All domains include a substantial number of OOFS items, preventing inflated performance due to repeated fingerspelling strings across splits.

**Visual stratification (Lab).** The Lab subset contains synchronized recordings from multiple viewpoints. Even when lexical items coincide across splits, differences in viewing angle and spatial configuration reduce the risk of pixel-level memorization and encourage viewpoint-robust evaluation.

**Character distribution.** We report full letter-frequency statistics in Table 5 of the supplementary material. Rare letters (e.g., "X", "J") appear in both training and test partitions, supporting fair assessment of long-tail character performance without artificially balancing the splits.

## C.3    AUSLAN-DAILY NEWS V2

Auslan experts manually annotate sentence-level temporal boundaries within the Auslan-Daily News videos to accurately align signed utterances with their corresponding English subtitles. This alignment process not only corrected mismatches between spoken captions and signing segments but also ensured each signed sentence was temporally grounded with high precision. As a result of this effort, we substantially expanded the original Auslan-Daily News subset and release it as a new version, termed **Auslan-Daily News V2**.

Table 6 presents a detailed comparison between the Auslan-Daily News V1 and V2 sub-datasets in terms of data volume, diversity, and vocabulary statistics. V2 significantly expands upon V1, featuring nearly twice the number of annotated segments (29,669 vs. 11,065), frames (5.6M vs. 2.3M), and total words (492,624 vs. 188,774). It also includes a larger vocabulary size (15,976 vs. 12,346) and more signers (27 vs. 18), reflecting improved linguistic and signer diversity. The increase in out-of-vocabulary (OOV) words and singletons further illustrates the dataset's long-tail lexical

Table 6: Key statistics of Auslan-Daily New V1 and Auslan-Daily New V2. OOV: out-of-vocabulary. Singleton: words that only occur once in the training dataset.

| Sub-Dataset | Auslan-Daily News V1 (Shen et al., 2023) | | | Auslan-Daily News V2 | | | |
|---|---|---|---|---|---|---|---|
| Domain/Topic | News & Documentary | | | News | | | |
| Video Resolution@FPS | 1280×720/1920×1080@29.97 | | | 1280×720@25 | | | |
| Split | Train | Dev | Test | Train | Dev | Test | Total |
| Segments | 9,665 | 700 | 700 | 16,604 | 1,000 | 1,000 | 29,669 |
| Signers | 18 | 17 | 17 | 24 | 22 | 19 | 27 |
| Frames | 2,072,475 | 144,819 | 142,893 | 2,925,597 | 157,619 | 149,984 | 5,593,387 |
| Vocab. | 12,346 | 2,872 | 2,885 | 13,767 | 3,020 | 3,010 | 15,976 |
| Tot. words | 163,268 | 11,376 | 11,530 | 277,699 | 14,343 | 14,408 | 492,624 |
| Tot. OOVs | - | 326 | 304 | - | 217 | 224 | 475 |
| Singletons | 5,267 | - | - | 6,110 | - | - | 8,039 |

Table 7: Examples of fingerspelling (FS) instances and their corresponding aligned content.

| Sentence | Fingerspelled Sequence | Aligned Text |
|---|---|---|
| The growth is going to have to rely heavily on **equity** students wanting to go to university. | E Q T I T Y | equity |
| We've used Variety in the past for some of his **equipment** and support. | E Q | equipment |
| The Western Bulldogs beaten **Greater Western Sydney**. | G W S | Greater Western Sydney |
| Irishwoman Leona **Maguire** has a one-shot lead heading. | M A Q U I R E | Maguire |
| Steven **miles** did a good job as leader. | M I I M I L E S | miles |
| He said he will get a job. | B O B | No aligned word |

distribution, which poses challenges but also fosters better generalization in sign language translation and recognition models.

## C.4 FINGERSPELLING ANNOTATION GUIDELINES.

To ensure consistency and accuracy in fingerspelling (FS) labeling, we adopt a structured three-step annotation protocol:

1. **Temporal Identification:** Annotators first review the entire video and identify all time intervals where fingerspelling occurs. These segments are typically characterized by rapid handshapes corresponding to individual alphabet letters, often used to spell out names, technical terms, or out-of-vocabulary words.

2. **Character-Level Transcription:** Within each identified FS segment, annotators transcribe the fingerspelled content into a sequence of characters (A–Z), ensuring the character sequence reflects the exact order and repetition observed in the signing. Ambiguous or occluded handshapes may be annotated with a special token (e.g., '*') when necessary.

3. **English Alignment:** After obtaining the character-level transcription, annotators check whether the transcribed fingerspelling sequence corresponds to any English word or phrase within the aligned sentence. If a match is found, the FS sequence is linked to the corresponding word or phrase. If no such alignment exists (e.g., due to fingerspelling of foreign names or uncommon entities), the segment is annotated as *not aligned* with any English content.

As shown in Table 7, we present several representative samples from our fingerspelling annotation process. Each example includes the full sentence containing a fingerspelled segment, the transcribed character sequence, and the corresponding aligned English word or phrase when available. This protocol ensures that each FS annotation is temporally precise, linguistically grounded, and aligned with surrounding sentence context where possible, enabling reliable training of FS recognition and translation models.

## C.5 FINAL DATASET STORAGE

The finalized BANZ-FS dataset, curated and annotated by sign language experts and trained annotators, is structured into three task-specific subfolders and is hosted on a public cloud repository. The organization of the dataset is illustrated in Figure 7. Each subfolder corresponds to a specific task: *Fingerspelling Recognition*, *Fingerspelling Detection*, and *Sign Language Translation*, respectively.

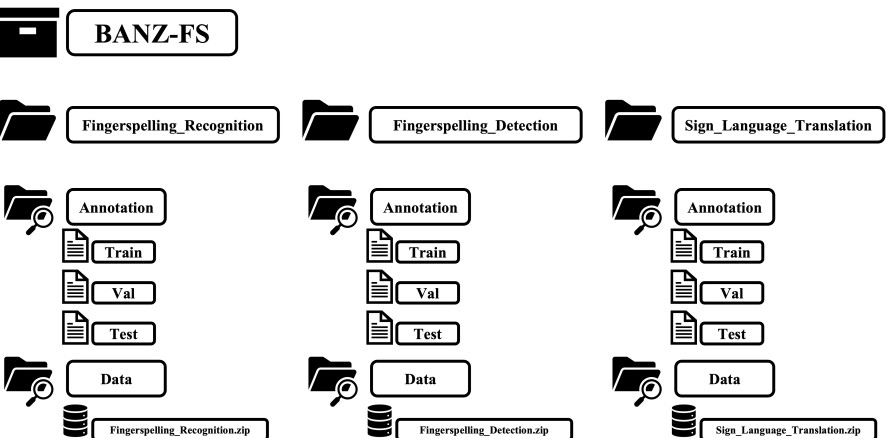

Figure 7: Hierarchical data folders for BANZ-FS on **G** Google Drive.

Table 8: Statistics of isolated versus continuous fingerspelling clips. "Length" refers to total duration in seconds; "Average" is the mean duration per clip.

| Source | Isolated FS Clips | | | Continuous FS Clips | | |
|---|---|---|---|---|---|---|
| | #Clips | Length | Average | #Clips | Length | Average |
| ABC News | 22,198 | 17,483.39 | 0.79 | 3,678 | 6,471.37 | 1.76 |
| YouTube | 2,149 | 5,770.38 | 2.69 | 374 | 2,336.38 | 6.25 |
| Lab Recordings | 10,732 | 14,141.18 | 1.31 | 0 | 0.00 | 0.00 |

Within each task folder, the dataset is further divided into two main components. The *Annotation* subfolder contains three files: `Train`, `Val`, and `Test`, each storing task-specific labels such as sentence-level alignments, fingerspelling boundaries, or character sequences depending on the task. These annotations are derived from expert manual alignment procedures and reflect high-quality temporal labeling. The *Data* subfolder contains a compressed archive (e.g., `Fingerspelling_Recognition.zip`) holding all the corresponding RGB video clips. These videos are already pre-processed to focus on the signer and are trimmed according to the annotated segment durations.

This storage structure ensures modular access for each task, allowing researchers to independently work on detection, recognition, or translation without ambiguity. Additionally, all annotations are time-aligned with video content, facilitating temporal learning and evaluation. An overview of the folder structure is shown in Figure 7, and recommended splits are discussed in Table 2 and Table 6.

## C.6 CONTINUOUS FINGERSPELLING CLIPS

In our annotations, each lexical fingerspelled item is treated as the minimal unit. For example, in the utterance "Here are the forecasts for Brisbane, New South Wales, and Sydney", although the FS segments "BB", "NSW", and "SY" appear consecutively, we annotate each of them as a separate fingerspelling segment with its own time interval. This segmentation strategy results in an average clip length of approximately 1.5 seconds across the dataset.

To support research on long-form fingerspelling detection and recognition, we additionally merge temporally adjacent fingerspelling segments and report statistics for these continuous spans. The statistics are summarized in Table 8, and both isolated and continuous segment boundaries will be included in the public release to enable future benchmarking efforts.

These findings highlight the linguistic and stylistic diversity captured in BANZ-FS: fingerspelling in news broadcasts tends to be fast and compressed, whereas YouTube content is slower and more naturalistic. Although our Lab Recordings subset does not include continuous FS clips, it contains fine-grained temporal annotations for each lexical item, enabling generation of high-quality synthetic continuous sequences in future work.

Table 9: Signer Demographic Breakdown by Subset.

| Subset | #Signers | Gender (M/F) | Age Range | Region | Demographics (Caucasian / Asian / African) |
|---|---|---|---|---|---|
| Lab Recordings | 67 | 38 / 29 | 16–75 | BANZ | 27 / 30 / 10 |
| News Clips | 29 | 11 / 18 | 25–55 | Auslan | 23 / 6 / 0 |
| Web Data | 20 | 7 / 13 | 20–45 | BSL, NZSL | 16 / 2 / 2 |

## C.7 DEMOGRAPHIC REPRESENTATION AND COVERAGE

As shown in Table 9, we provide a demographic summary of the signers in our dataset, including gender, age range, collection region, and demographics across the three subsets: Lab Recordings, News Clips, and Web Data. While Web and News subsets rely on public sources and offer limited control over signer demographics, we address this by proactively recruiting a diverse signer pool in the Lab Recordings subset, ensuring broader representation in terms of age and racial background. We also confirm that our collection covers participants from multiple regions within Australia, and the Web subset includes samples from the UK and New Zealand. Although Auslan content makes up the majority of the dataset, the BANZSL fingerspelling alphabet is shared across all dialects, providing a solid foundation for cross-dialect generalization. We acknowledge that perfect demographic balance (Section B) is difficult to achieve, but our ongoing efforts in data collection and documentation aim to support transparent and inclusive dataset construction.

## D CONSENT FORM FOR BANZ-FS RECORDING

**Consent Form for Recording of the Australian Sign Language Dataset**

Dear Participant,

Hello! We are a team dedicated to the research of sign language. We are conducting an academic project aimed at recording and analyzing Australian Sign Language (Auslan). We invite you to participate in this project. The purpose of this project is to facilitate the learning and dissemination of sign language and to enhance understanding and application of Auslan.

**Mode of Participation:**
You will be recorded while using Auslan for communication. These recordings may include your facial expressions and hand gestures.

**Privacy and Data Use:**
We commit to using the recorded data solely for academic research purposes and not for any commercial use. All data will be anonymized to ensure the security of your personal information. The video material may be presented at academic conferences, in research papers, or educational courses.

**Consent Details:**

1. I have read and understood the information about the research described above.
2. I agree to participate in the video recordings of Australian Sign Language.
3. I understand that my participation is voluntary, and I can withdraw at any time without any adverse consequences.
4. I agree that my facial expressions and hand gestures may be recorded and used for academic research.

Please fill out the following information and sign below to indicate your consent to participate:

- **Name:** _______________________
- **Email:** _______________________
- **Signature:** _______________________
- **Date:** _______________________

We greatly appreciate your participation and support!

Should you have any questions or require further information, please contact us at:

**Contact Person:** [Name of Coordinator]
**Email:** [Coordinator's Email]
**Phone:** [Coordinator's Phone]

Figure 8: Consent Form for Recording.

Due to the inclusion of facial information in our dataset, we obtain consent from volunteers and have them sign the consent form depicted in Figure 8 before recording data. **We do not release personally identifiable information** such as names, ages, occupations, or indications of whether individuals are deaf or hard of hearing. It is important to note that our dataset is strictly for academic use and can not be used for commercial purposes.

## E  MORE DETAILS FOR VIDEO REPRESENTATION

**RGB-based:**  We use the pre-trained I3D model form (Li et al., 2020a) and features with a window width of 16 and a stride of 2 are extracted:

$$f_t = \text{I3D}(F_{t-\frac{n}{2}} \oplus ... \oplus F_t \oplus ... \oplus F_{t+\frac{n}{2}}), \quad (1)$$

where $f_t$ is the representation of the $t$-th frame, $n$ is the window width, and $\oplus$ denotes the concatenation operation.

**Pose-based:**  Leveraging pose information in action recognition presents significant benefits regarding robustness and semantic representation. We flatten the pose array $A \in R^{T \times N \times 2}$ to $A_f \in R^{T \times 2N}$, where $T$ is the number of frames and $N$ is the number of keypoints. Meanwhile, the results of our experiment show that the use of partial body and two hand keypoints will perform better for tasks related to sign language (Xu et al., 2025; Shen et al., 2025c).

## F  EXPERIMENTAL SETTINGS

We mention that all models used in this work are publicly available. Each of the models we use is linked below:

- **Isolated Fingerspelling Recognition:**
  SL-Transformer (Camgöz et al., 2020) ⭯, Iterative-Att (Shi et al., 2019) ⭯, MiCT-RANet (Mahoudeau, 2020) ⭯, TS-FS-Reg (Chen et al., 2022c) ⭯ and FS-PoseNet (Fayyazsanavi et al., 2024) ⭯.

- **Fingerspelling Detection:**
  Bi-LSTM CTC (Huang et al., 2015) ⭯, Modified R-C3D (Xu et al., 2017) ⭯, TS-FS-Det (Chen et al., 2022c) ⭯, MT-FS-Det (Shi et al., 2021) ⭯, and SL-Seg (Moryossef et al., 2023) ⭯.

- **Fingerspelling Recognition in Context and Sign Language Translation:**
  SL-Luong (Luong et al., 2015) ⭯, SL-Transf (Camgöz et al., 2020) ⭯, TSPNet (Li et al., 2020b) ⭯, MMTLB (Chen et al., 2022a) ⭯, GASLT (Yin et al., 2023) ⭯, and GFSLT-VLP (Zhou et al., 2023) ⭯.

We express profound gratitude to the aforementioned authors for their invaluable contributions.

All the training and fine-tuning experiments are run on a machine with four NVIDIA GeForce RTX 3090 GPUs. We use the default hyperparameters for training the models of fingerspelling-related tasks and sign language translation.

## G  THE BASELINE OF AUSLAN-DAILY NEWS V2

To evaluate the effectiveness of existing sign language translation (SLT) models on our extended dataset, we benchmark several state-of-the-art gloss-free SLT systems on both Auslan-Daily News V1 (Shen et al., 2023) and our newly constructed Auslan-Daily News V2. We consider two settings based on input pre-processing:

- **Single-Person SLT:** The signer is automatically detected and cropped from the original video. This setting eliminates most background noise and visually isolates the signing individual.

- **Multi-Person SLT:** The entire video frame is preserved, including other people and background elements. Although only one person performs sign language in these clips, the presence of scene context and distractors makes translation more challenging.

Table 10: Translation results of Single/Multi-Person SLT gloss-free models on Auslan-Daily News (Shen et al., 2023) and our newly extend Auslan-Daily News V2.

| | | Auslan-Daily News V1 (Shen et al., 2023) | | | | | Auslan-Daily News V2 | | | | |
|---|---|---|---|---|---|---|---|---|---|---|---|
| Single-Per. SLT | Input | R | B1 | B2 | B3 | B4 | R | B1 | B2 | B3 | B4 |
| SL-Luong (Luong et al., 2015) | Pose | 20.65 | 19.84 | 7.81 | 4.59 | 2.81 | 21.71 | 21.42 | 10.55 | 6.58 | 4.94 |
| SL-Luong (Luong et al., 2015) | RGB | 16.14 | 16.92 | 7.44 | 4.07 | 2.68 | 15.47 | 16.43 | 7.61 | 5.19 | 3.91 |
| SL-Transf (Camgöz et al., 2020) | Pose | 20.25 | 21.25 | 6.57 | 3.32 | 2.11 | 22.17 | 21.61 | 8.12 | 4.84 | 3.65 |
| SL-Transf (Camgöz et al., 2020) | RGB | 14.93 | 17.64 | 7.41 | 3.98 | 2.52 | 13.83 | 13.93 | 7.02 | 4.33 | 3.05 |
| TSPNet-Joint (Li et al., 2020b) | RGB | 19.71 | 18.23 | 5.97 | 3.21 | 2.26 | 20.09 | 20.39 | 7.66 | 4.23 | 2.83 |
| MMTLB (Chen et al., 2022a) | RGB | 18.90 | 19.64 | 5.30 | 3.26 | 2.31 | 16.80 | 18.80 | 8.01 | 5.15 | 3.68 |
| GASLT (Yin et al., 2023) | Pose | 18.76 | 15.57 | 6.06 | 3.72 | 2.72 | 24.78 | 21.43 | 9.91 | 6.19 | 4.26 |
| GASLT (Yin et al., 2023) | RGB | 22.01 | 19.54 | 7.45 | 4.41 | 2.56 | 23.94 | 20.99 | 8.22 | 6.08 | 3.77 |
| GFSLT-VLP (Yin et al., 2023) | RGB | **27.32** | **23.00** | **9.93** | **6.08** | **4.43** | **26.24** | **22.59** | **11.50** | **7.29** | **5.44** |
| Multi-Per. SLT | Input | R | B1 | B2 | B3 | B4 | R | B1 | B2 | B3 | B4 |
| SL-Luong (Luong et al., 2015) | RGB | 14.04 | 15.53 | 6.11 | 3.27 | 2.05 | 13.61 | 14.14 | 6.62 | 4.67 | 3.30 |
| SL-Transf (Camgöz et al., 2020) | RGB | 13.68 | 16.58 | 5.86 | 2.72 | 1.55 | 15.05 | 17.29 | 7.63 | 4.49 | 3.09 |
| TSPNet-Joint (Li et al., 2020b) | RGB | 14.64 | 17.33 | 3.86 | 1.66 | 1.89 | 14.68 | 17.77 | 6.90 | 3.69 | 2.33 |
| MMTLB (Chen et al., 2022a) | RGB | 17.76 | 16.02 | 4.81 | 2.83 | 1.83 | 20.69 | 21.21 | 7.07 | 3.67 | 2.43 |
| GASLT (Yin et al., 2023) | RGB | 19.73 | 16.99 | 6.25 | 3.44 | 2.26 | 20.78 | 19.43 | 6.91 | 4.79 | 3.56 |
| GFSLT-VLP (Yin et al., 2023) | RGB | **20.83** | **18.93** | **6.02** | **4.33** | **3.05** | **21.87** | **18.07** | **7.66** | **5.12** | **4.10** |

We evaluate models using RGB and pose input modalities, reporting BLEU scores (B1–B4) and ROUGE (R) metrics in Table 10. The results clearly show that Auslan-Daily News V2 is more challenging than V1, with slightly lower scores across all models and metrics, especially under the Multi-Person setting. This highlights the increased variability and complexity introduced by our new annotations and broader content coverage.

Among the tested models, GFSLT-VLP (Zhou et al., 2023) consistently achieves the best performance across both datasets and settings, demonstrating the benefit of vision-language pretraining. Notably, Single-Person setups tend to outperform Multi-Person ones, confirming that signer isolation reduces visual ambiguity and aids translation. These baselines provide strong references for future research on realistic, scalable, and context-aware SLT in broadcast news environments.

## H  ADDITIONAL DISCUSSION

### H.1  EMPIRICAL JUSTIFICATION FOR TEMPORAL SEGMENTATION DESIGN

In the data construction phase, we adopt a fixed 10-second sliding window strategy for segmenting fingerspelling sequences. Here, we provide the rationale and empirical validation for this design choice from three perspectives: sequence coverage, training robustness, and performance comparison.

**Coverage of Long-tail Durations.** Fingerspelling sequences in BANZ-FS exhibit significant variance in duration across different domains. While many isolated segments in the Lab subset are relatively short ($< 1.5s$), naturalistic continuous fingerspelling in the Web subset is significantly longer. As detailed in Table 8 8, continuous clips from YouTube have an average duration of **6.25 seconds**, with some instances significantly exceeding this length. A shorter window (e.g., 5 seconds) would pose a high risk of truncating these naturalistic, long-tail sequences, leading to the loss of critical start/end tokens and visual context.

**Robustness via Temporal Randomness.** Our sliding window approach is designed to capture a 10-second context around any detected fingerspelling segment without strictly centering it. This design allows the target sequence to appear at variable temporal positions within the window. This introduces an implicit form of *temporal data augmentation* during training. By exposing the model to varying temporal offsets, we encourage the learning of translation-invariant features, thereby improving robustness against the imprecise temporal proposals often encountered in real-world detection scenarios.

Table 11: Ablation study on sliding window size for Fingerspelling Detection (AP@IoU$_{0.5}$).

| Window Size | News | Lab | Web | Full |
|---|---|---|---|---|
| 5 seconds | 53.4 | 82.7 | 45.9 | 66.2 |
| 10 seconds | **53.9** | **82.7** | **47.3** | **66.9** |

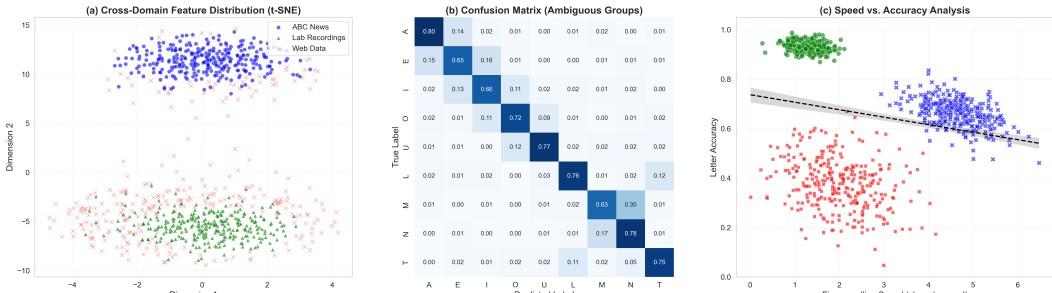

Figure 9: **Comprehensive Analysis of Domain Variability and Error Sources. (a) Cross-Domain Feature Distribution:** t-SNE visualization of video features reveals distinct clusters for News (Blue) and Lab (Green), while Web data (Red) exhibits high variance, confirming the domain gap. **(b) Confusion Matrix:** Highlights specific visual ambiguities. **(c) Speed vs. Accuracy:** Empirically quantifies the negative correlation between signing speed and recognition accuracy, identifying rapid transitions in News data as a key bottleneck.

**Ablation Study: 5s vs. 10s Window.** To empirically validate the impact of window size, we conducted an ablation study comparing the performance of the baseline detection model (SL-Seg) trained with a 5-second window versus our proposed 10-second window. As observed in Table 11, the 10-second window yields consistently superior performance compared to the 5-second setting. The performance degradation observed with the 5-second window confirms that the larger temporal context is essential for handling variable-length, in-the-wild data, while simultaneously avoiding truncation errors.

## H.2 Visualization of Cross-Domain Feature Distribution

To qualitatively analyze the domain variability and generalization challenges discussed in Section 5.2, we conducted a t-SNE (Maaten & Hinton, 2008) visualization of the video feature representations extracted from our three data sources: ABC News, Lab Recordings, and Web Data. We utilized the same HandReader (RGB) (Korotaev et al., 2025) backbone used in our benchmark experiments to extract high-dimensional features from a randomly sampled subset of video clips, which were then projected into a 2D space.

As illustrated in Figure 9(a), the feature distributions exhibit clear patterns that corroborate our quantitative findings. Specifically, the **ABC News** subset (Blue) and **Lab Recordings** (Green) form distinct, tightly grouped clusters, reflecting the standardized studio environment and professional lighting typical of these controlled settings. In stark contrast, the **Web Data** (Red) exhibits a highly dispersed distribution with significantly larger variance; notably, its manifold spatially surrounds and partially overlaps with the Lab cluster but extends into sparse regions of the feature space. This visualizes the "in-the-wild" nature of the Web subset, encompassing a wide spectrum of visual conditions from simple setups to highly complex scenarios. This high variance visually justifies the performance drop observed in cross-domain benchmarks and underscores the necessity of diverse training data for robust fingerspelling recognition.

## H.3 Quantitative Error Analysis

To complement the qualitative case studies in Section 5.3 and provide deeper diagnostic insights into model failures, we conducted two quantitative analyses: per-letter confusion analysis and speed-performance correlation, as illustrated in Figure 9.

As shown in Figure 9(b), the confusion matrix verifies distinct error patterns specific to the two-handed BANZSL system. A significant source of error arises within the consonant group, particularly between 'M' and 'N'. This misclassification is attributed to the subtle visual difference in finger counting (three vs. two fingers), which is easily obscured by motion blur or self-occlusion during continuous signing. Additionally, we observe mutual confusion among the vowel group ('A', 'E', 'I', 'O', 'U'). Since BANZSL vowels involve contacting specific fingertips of the non-dominant hand, rapid articulation often leads to imprecise contact localization, causing the model to struggle in distinguishing these spatially adjacent classes.

To analyze the impact of temporal dynamics, Figure 9(c) plots Letter Accuracy against Fingerspelling Speed. A clear negative correlation is observed: as signing speed increases, recognition accuracy consistently declines. The data distribution reflects domain-specific challenges. The **Lab** recordings cluster in the low-speed and high-accuracy region, serving as an upper bound. In contrast, the **News** domain is characterized by high speed and moderate accuracy, confirming that rapid inter-letter transitions significantly degrade performance for professional signers. Finally, the **Web** domain suffers from lower accuracy regardless of speed, indicating that visual factors such as lighting and background clutter, rather than speed alone, are the dominant sources of error in in-the-wild settings.

### H.4 STRATIFIED ANALYSIS OF FACTORS AFFECTING FINGERSPELLING PERFORMANCE

To better understand which factors influence fingerspelling detection and recognition on BANZ-FS, we conduct stratified analyses across dialect, signer profile, and letter frequency.

**Dialect and context.** We first examine performance across the three BANZSL dialects. The BSL/NZSL-dominant Web subset is more challenging than the Auslan-dominant News and Lab subsets, which is consistent with its smaller annotated scale and greater contextual variation. Training on the combined *Full* dataset improves performance on the Web test set relative to training on Web-only data (Table 3), suggesting that enlarging the multilingual training pool benefits cross-dialect generalization.

**Signer profiles.** We further analyze performance by signer type and signing speed. As reported in Table 2, News interpreters fingerspell substantially faster (4.59 characters per second) than Lab and Web signers (approximately 1.30 characters per second). Correspondingly, models achieve the highest accuracy on the slower Lab data and the lowest on the rapid, highly fluent News data.

**Letter frequency.** To analyze how letter frequency affects recognition accuracy, we stratify performance by individual characters using the HandReader model trained on the *Full* dataset. Table 5 reports the accuracy of the most and least frequent letters. Frequent characters such as "N" and "A" reach approximately 83% and 80% accuracy, respectively, whereas rare characters such as "X" and "J" achieve around 54% and 41%. These results demonstrate a clear correlation between letter occurrence and recognition accuracy and highlight the long-tail challenge in BANZ-FS. Future extensions of the dataset will explicitly target low-frequency characters to improve model robustness.

## I CASE STUDY FOR BANZ-FS FINGERSPELLING

Figure 10 illustrates qualitative examples from our BANZ-FS dataset, comparing ground truth annotations with predictions from three systems. Across most examples, FSD-IFSR demonstrates accurate segmentation and character-level recognition, closely matching the ground truth, especially in clear and isolated contexts. However, recognition becomes more challenging in broadcast news settings, as shown in the top-right example. The model confuses the final "N" with "M", likely due to their similar handshapes and coarticulation under fast signing. This highlights the visual ambiguity of adjacent characters in rapid sequences. Another common error arises from temporal proximity of repeated letters. In the bottom-right example, the system fails to distinguish the two or three "0"s near the end, merging them into a single instance. This suggests the need for improved temporal modeling to separate closely spaced, visually similar gestures.

## J CASE STUDY FOR AUSLAN-DAILY NEWS SIGN LANGUAGE TRANSLATION

Table 12 showcases qualitative examples from our Auslan-Daily News V2 translation benchmark. We compare ground-truth sentences with outputs from SL-Luong + Pose (Camgöz et al., 2020) and GFSLT-VLP (Zhou et al., 2023) models. Both models perform well on short, common sentences, but longer or more complex utterances reveal clear differences. GFSLT-VLP captures more complete sentence structures and preserves key semantic information better than the baseline.

## K LLM USAGE STATEMENT

Large Language Models (LLMs) such as ChatGPT are used as general-purpose tools to improve readability and clarity of the manuscript, e.g., for grammar checking, LaTeX formatting, and restruc-

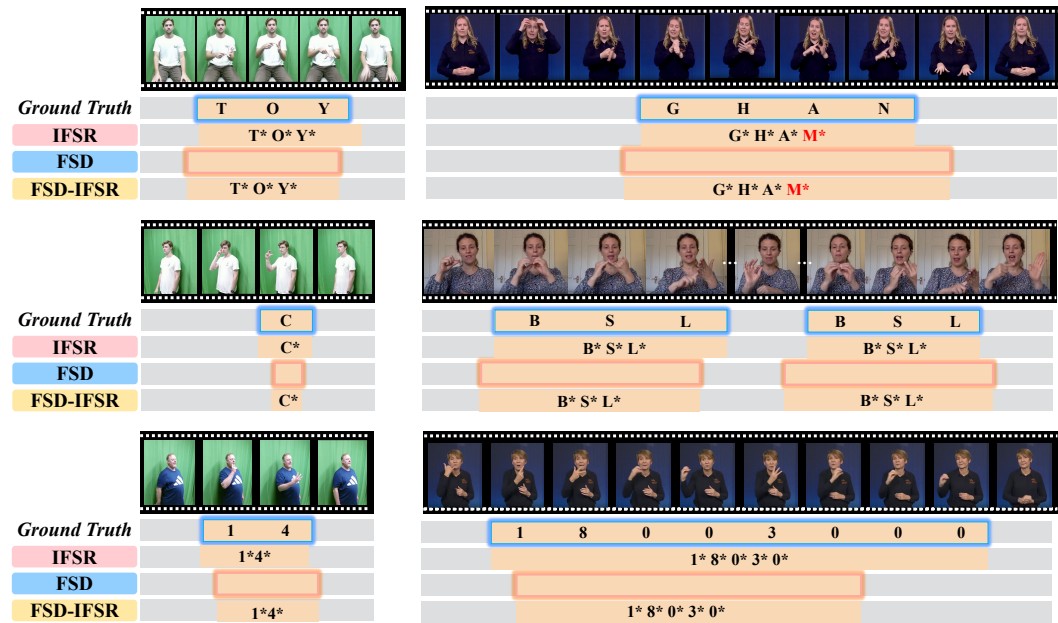

Figure 10: Case study comparing IFSR, FSD, and FSD-R on fingerspelling sequences.

Table 12: Case study. We highlight exactly correct translations in red and missing contents in blue.

| GT | hello and welcome to abc news . |
|---|---|
| SL-Luong + Pose (Camgöz et al., 2020) | hello and welcome to abc news . |
| GFSLT-VLP (Zhou et al., 2023) | hello and welcome to abc news . |
| **GT** | **do you think thing have significantly change in the last year .** |
| SL-Luong + Pose (Camgöz et al., 2020) | do you think there be escalate ... |
| GFSLT-VLP (Zhou et al., 2023) | do you think thing have significantly change in the last year . |
| **GT** | **to new south wale and the act rain and cool in the northeast .** |
| SL-Luong + Pose (Camgöz et al., 2020) | rain and cool in the east . |
| GFSLT-VLP (Zhou et al., 2023) | to new south wale and the act rain and cool in the northeast . |
| **GT** | **the prime minister have reject suggestion he redefine his ...** |
| SL-Luong + Pose (Camgöz et al., 2020) | prime minister have a new suggestion ... |
| GFSLT-VLP (Zhou et al., 2023) | the prime minister have reject suggestion that he ... |

turing sentences. No parts of the research idea, dataset design, or experimental results are generated or influenced by LLMs. All technical contributions and conclusions are solely those of the authors.

