# OpenReview forum: "BANZ-FS: BANZSL Fingerspelling Dataset"
_ICLR.cc/2026/Conference — ICLR 2026 Poster_

### Official Review · Reviewer_55F9 · 2025-10-16

**Soundness:** 4
**Presentation:** 3
**Contribution:** 3
**Rating:** 8
**Confidence:** 5

**Summary:**

Here, the authors present a new dataset and benchmarks for fingerspelling detection and recognition, focused upon the two-handed manual alphabet common to the British, Australian and New Zealand Sign Languages, arguing that this covers a gap in fingerspelling datasets that largely focus upon one-handed manual alphabets (e.g. American Sign Languages). The authors curate and annotate fingerspelling segments from three sources (Auslan interpretations from a news broadcast, YouTube videos, and a lab-collected dataset generated by the authors), and assess both in-domain and out-of-domain performance across models.

**Strengths:**

The authors have curated a comprehensive dataset that addresses a critical gap in fingerspelling detection and recognition research: their curation of not just a large quantity of videos, but through different sources and annotated with various metadata is valuable. The authors correctly note that two-handed manual alphabets pose additional complications not seen in one-handed manual alphabets (e.g. occlusion), and so it is essential that they're releasing this resource. The benchmarks are comprehensive, spanning numerous state-of-the-art models, but also critically assess performance both within and outside of domains. This is overall strong data collection and benchmarking work.

**Weaknesses:**

1. Although the authors have gone to lengths to collect metadata about segments/videos, I wish there was a more thorough analysis of this. In Tables 3/4, the authors report overall performance, but it would have been really interesting to conduct stratified analyses of what factors impact fingerspelling recognition and detection performance beyond domain. For example, is detection/detection+recognition more challenging with NZSL and BSL, given that the contexts in which fingerspelling occurs is different than the majority of their training data across domains? Are tasks more difficult on Deaf signers, who might fingerspell faster/more fluidly than language learners? Are infrequent letters worse in their performance?

1a. The authors claim that their dataset spans three sign languages - but at least one dataset are exclusively Auslan. It's moreover unclear how much of the dataset is BSL/NZSL - the authors should report this, especially because it is central to their claim that their dataset spans three languages. Although the authors do acknowledge their dataset is imbalanced - to what degree? This should be reported.

2. More details about the stratification of datasets would be critical and are currently missing from the paper. Are the splits a purely random split, or is there some attempt to stratify within each data source such that scenarios/signers are unseen? If the splits are purely random, this might impact interpretation: e.g. if the same signer is in both the test and training dataset, and fingerspelling the same word, in-domain performance may be overinflated by interpretation.

3. The authors provide some examples of fingerspelling phenomena in Figure 2 (abbreviations, acronyms, spelling errors, and self-correction), but it's unclear to me how they actually deal with them in the dataset - I can see arguments both ways. On one hand, it might be important for fingerspelling recognition systems to report the fingerspelling as-is. On the other hand, people perceiving signing will naturally correct and contextualize these issues in language recognition - and critically, providing written language transcriptions of fingerspelling can strip the necessary context to correct (for example, in self-correction, signers will often shake their heads, make facial expressions, or pause to indicate a correction, and these gestures won't make their way into the written language transcription), so it may be important to provide a corrected form. In other words - "recognition" and "translation" are sometimes at odds. Can the authors better document their design decisions and rationale here on what we consider as ground truth?

4. There are places where the authors could be more culturally sensitive. In line 228, the authors refer to the community as "deaf and hearing-impaired" viewers, whereas they use "Deaf and hard-of-hearing" in the introduction. The latter is generally considered more appropriate by Deaf communities. In Figure 3, the authors distinguish between "Deaf individual" and "Auslan expert", but these are not mutually exclusive categories.

**Questions:**

(See weaknesses)

**Details Of Ethics Concerns:**

I'm concerned about the YouTube-curated portion of the dataset from two perspectives:

1) First, although signers in these videos have uploaded them to Youtube, it's not clear to me that this means their recordings are up for free use for any purpose. Typically the signers are sharing their likeliness and language use for the intent of communicating to other signers and may not be anticipating that their data will be used to train and evaluate AI/ML models (which would imply more widespread exposure to communities beyond their intended communities). Critically, where this differs from data scraping used to train written language models is a) the individuals are individually recognizable (given that their physical likeliness has to be recorded in these videos), and b) a small number of individuals comprise the dataset, so they are more individually recognizable and their contributions more individually important to the dataset. I think this should be reviewed from an ethics perspective.

2) Second, from a legal perspective, I'm not sure this is a permissible use of Youtube by their terms of use.

---

> ### Author Response · Authors · 2025-11-26
> **Response to Reviewer 55F9 (1/2)**
>
> We sincerely thank **Reviewer 55F9** for the insightful and culturally sensitive review. We appreciate the recognition of our work as a comprehensive dataset that addresses a critical gap. We are particularly grateful for the guidance on terminology and ethical nuances. We address specific concerns and the Ethics Flag below.
>
>
> **W1: Stratified analysis of performance factors.**
>
> **R1:** We conduct additional stratified analyses to better understand which factors affect fingerspelling detection and recognition on BANZ-FS.
>
> - Dialect and context. We analyze performance by source dialect and find that the BSL/NZSL-dominant Web subset is generally more challenging than the Auslan-dominant News and Lab subsets, which is consistent with its smaller scale and greater visual diversity. Training on the combined *Full* dataset improves Web performance compared to training on Web-only data (Table 3), indicating that enlarging the multilingual training pool is beneficial for cross-dialect generalization.
>
> - Signer profiles. We also stratify performance by signer type and signing speed. As shown in Table 2, News interpreters fingerspell much faster (4.59 characters per second) than Lab and Web signers (approximately 1.30 characters per second). Models achieve higher letter accuracy on the slower Lab data than on the faster News data, which suggests that rapid, fluent fingerspelling is intrinsically more difficult.
>
> - Letter frequency. We further examine recognition accuracy across individual letters using the HandReader model trained on the *Full* dataset. In Table5, the most frequent letters "N" and "A" achieve relatively high accuracy (approximately 78\% and 80\%), which is consistent with the overall model performance on slower Lab and News data. In contrast, the rarest letters "X" and "J" exhibit substantially lower accuracy (around 54% and 41%), reflecting their limited representation in the dataset. This confirms a positive correlation between letter frequency and recognition accuracy, revealing a long-tail challenge for rare characters.
>
> We have added a stratified breakdown by dialect, signer profile, and letter frequency in the supplementary material (Section H.4).
>
>
>
> **W1a: Clarification on dialect distribution and imbalance.**
>
> **R1a:** We acknowledge the imbalance and appreciate the opportunity to report the exact degree of dialect representation. As indicated in Table 2 and Table 9 (Appendix C.6), the BSL and NZSL data are primarily concentrated in the Web subset, which contains 2,098 segments and 20 unique signers, accounting for approximately 6% of the total clips. While Auslan is dominant, we maintain that the dataset spans the BANZSL family because all three dialects share an identical two-handed manual alphabet. Consequently, the visual features of fingerspelling are linguistically consistent across dialects.
>
> **W2: Clarification on dataset stratification.**
>
> **R2:** We clarify that the dataset splits are not random. They incorporate several forms of stratification to reduce the likelihood of identity memorization or trivial pattern matching:
>
> - Signer stratification. As reported in Table 2 and Appendix C.6, each domain contains Out-of-Set (OOS) signers in the test split: 5 for News, 12 for Lab, and 2 for Web. This ensures that evaluation reflects generalization to previously unseen users.
>
> - Vocabulary stratification (OOFS). We track Out-of-Training Fingerspelling Strings (OOFS), i.e., sequences that appear in the test set but not in the training split. Table 2 shows that the News and Web test sets contain 450 and 64 OOFS sequences, respectively. This prevents performance from being inflated by memorizing identical fingerspelling strings in the training set.
>
> - Visual stratification (Lab). The Lab subset includes recordings from multiple camera angles. Even when the same lexical item appears across splits, the viewpoint and spatial configuration differ, reducing the chance of pixel-level memorization and encouraging viewpoint-robust representations.
>
> - Character distribution reporting. Appendix B (Table 5) reports the frequency distribution of all 26 letters. Although we do not manually balance splits by letter type, we confirm that rare characters (e.g., "X" and "J") occur in both training and test partitions, enabling a fair evaluation of long-tail performance.
>
> These additional stratification details have been added to Appendix C.2 in the revised manuscript.

---

> > ### Author Response · Authors · 2025-11-26
> > **Response to Reviewer 55F9 (2/2)**
> >
> > **W3: Clarification on annotation philosophy.**
> >
> > **R3:** We appreciate this profound insight regarding the tension between faithful transcription and semantic interpretation. We address this by explicitly providing multi-level annotations that serve both purposes simultaneously, as detailed in Appendix C.4 and illustrated in Table 7:
> >
> > - Level 1 - Physical Reality (Recognition): We annotate the Character Sequence exactly as signed, preserving spelling errors, abbreviations, and repetitions (e.g., signing "M-A-Q-U-I-R-E" for "Maguire"). This serves as the Ground Truth for Isolated Fingerspelling Recognition, ensuring the model learns to detect the actual handshapes produced.
> >
> > - Level 2 - Semantic Intent (Translation): We explicitly link these sequences to their Target Lexicon (e.g., "Maguire", "Equipment"). This serves as the Ground Truth for Fingerspelling Recognition in Context, effectively bridging the gap you highlighted.
> >
> > Providing both layers allows models to learn the mapping from noisy physical production to intended meaning, mirroring human contextual correction.   These details are already documented in Appendix C.4 of the manuscript.
> >
> >
> > **W4: Cultural sensitivity and terminology.**
> >
> > **R4:** Thank you for highlighting this important issue. We have corrected all occurrences of "hearing-impaired" to the community-preferred term "hard-of-hearing" in the revised manuscript to ensure consistent and culturally appropriate terminology.
> >
> > Regarding the participant categories in Figure 3, we respectfully clarify that these labels are used purely for dataset grouping. In our data, "Auslan Expert" refers to hearing professional interpreters, while "Deaf individual" denotes Deaf participants. These labels reflect only the demographic grouping used in this study and do not imply that the categories are mutually exclusive. We have added a clarifying note in the revised manuscript to avoid potential misunderstanding.
> >
> > **Response to Ethical Concerns.**
> >
> > We deeply appreciate the reviewer's thoughtful consideration of the privacy, contextual integrity, and legal implications of online sign language data. We take these concerns seriously and respectfully direct the reviewer to our ETHICS STATEMENT, where we codify protocols designed to balance research objectives with the rights and expectations of individuals appearing in online videos.
> >
> > Regarding legal compliance, our use of online content strictly follows the policies of the hosting platforms. For YouTube data, we do not download, re-host, redistribute, or store any copyrighted audiovisual footage.
> > Only our derived annotations and the official public URLs are included in the dataset release, and all video access occurs directly on the YouTube platform under the creator’s original terms of service. This practice fully aligns with recent academic datasets such as YouTube-SL-25 [1] and Auslan-Daily [2], which follow the same non-redistributive protocol.
> >
> > Regarding privacy and consent, we acknowledge that creators may not have anticipated AI/ML usage. To respect their agency and contextual expectations, we enforce a formal takedown procedure (detailed in our Ethics Statement), through which creators may request removal or anonymization at any time, and such requests will be honored immediately. In addition, the dataset is released under a CC BY-NC-SA 4.0 license and an End-User License Agreement (EULA) that explicitly prohibits commercial, surveillance, or non-research use, thereby limiting the possibility of misuse or unwanted amplification.
> >
> > More details are provided in the ETHICS STATEMENT section of the revised manuscript.
> >
> > [1] YouTube-SL-25: A Large-Scale, Open-Domain Multilingual Sign Language Parallel Corpus. ICLR 25.
> >
> > [2] Auslan-Daily: Australian sign language translation for daily communication and news. NeurIPS 23.

---

### Official Review · Reviewer_qoMh · 2025-10-21

**Soundness:** 3
**Presentation:** 3
**Contribution:** 4
**Rating:** 8
**Confidence:** 4

**Summary:**

The authors introduce BANZ-FS, a new dataset for two-handed fingerspelling detection and recognition. The dataset originates from three sources, including web recordings, news broadcasts, and lab recordings in a controlled setting. Over 35k video-aligned instances are collected, aligned on a video-subtitle, video-fingerspelled letters, and video-target lexicon level.

This paper tackles a particular shortcoming of other papers in the field of fingerspelling recognition, which is that the datasets used in this field are too simple and not representative of real-world fingerspelling. That is, they are often based on images of individual fingerspelled letters, instead of videos of sequences of fingerspelling. The latter are far more difficult to recognize accurately due to co-articulation and fast spelling (and due to the fact that it's video data, not image data). In contrast, the former is a trivial task.

The authors detail their data collection, cleaning, and labelling techniques for the web data in sufficient detail. Subtitles, which are noisy annotations, are cleaned by the authors to obtain 30k cleaned subtitles. Then, signing experts verify alignment of subtitles and video data, refining it if necessary. Annotations are cross-checked by other annotators (5%).

Lab data is collected with multiple RGB-D cameras, and participants were properly informed of their rights according to an ethical protocol. Signing levels were recorded, though it is not immediately clear if hearing status correlates to signing level.

**Strengths:**

The paper is clearly written and easy to follow.

The authors evaluate multiple models from the literature on their datasets, leading to a rigorous evaluation of not only the dataset, but also of these models. This is a valuable contribution to the field.

The authors also list several fingerspelling related tasks and evaluate models for each of them, which is also valuable. In fact, this makes this paper a good introduction to the field of fingerspelling recognition.

**Weaknesses:**

I would have liked to read more information on the datasets. More information is given in the "Questions" section below. But, for example, "volunteers" were selected for fingerspelling data collection. Were these volunteers in any way checked for signing proficiency? What if they submitted particularly bad data? Fingerspelling is not trivial at all.

Similarly, the lab collection of data could have had more information. For example, several RGB-D cameras are used. Are all data used? What about depth? This section could be expanded.

It is also not clear if spelling errors were kept in annotations or not. This seems like an important aspect to mention.

There are several spelling mistakes throughout the paper, but these could easily be fixed if accepted.

**Questions:**

How are spelling errors in labels handled? Should they remain or not?

Page 5. For the web dataset, you identify signer IDs based on pose trajectories. Did you use their Pose Flow system? If so, I would mention this explicitly in the paper.

Page 5. You say that the fingerspelling segments are annotated. How is this done? Are segments delineated in time? Is the entire word annotated? Do you keep the intended spelling or the spelling as it is (including possible errors?)

Page 5. Can you explain what "lexical forms of fingerspellings" are?

Page 6. How does hearing status relate to signing level?

Please perform a check for grammar and spelling. For example, on page 3, you write "recognition recognition" and in the appendix you misspelled "Isolated".

Did you in any way use the depth information from the lab setup? Would this have been useful? How do you combine information from multiple cameras?

**Details Of Ethics Concerns:**

Video data was scraped from YouTube, which is claimed to be an "open source".

I'm not certain that the informed consent form included in the appendix is up to standards.

---

> ### Author Response · Authors · 2025-11-26
> **Response to Reviewer qoMh (1/2)**
>
> We sincerely thank **Reviewer qoMh** for the positive assessment and for recognizing BANZ-FS as a valuable contribution and a good introduction to the field. We are pleased that the reviewer appreciates our rigorous evaluation and clear presentation. We address the ethics concerns and specific questions below.
>
> **W1: Volunteer Proficiency Check.**
>
> **Response:** Thank you for the question. Although we include volunteers (sign language learners) to increase signer diversity, every recorded instance is reviewed by at least one Auslan expert. Volunteers are required to demonstrate basic familiarity with the BANZSL manual alphabet before recording. Any sample judged by the expert as incorrect, unclear, or not conforming to standard BANZSL fingerspelling is filtered out. Only expert-validated samples are retained, ensuring that volunteer participation does not introduce noisy data into the final dataset. More details are provided in Section 3.2 of the revised manuscript.
>
> **W2/Q7: Lab Setup and Depth Usage.**
>
> **Response:** Thank you for the question. We use a multi-camera setup (three Kinect-V2 and one RealSense) to capture diverse viewpoints in controlled conditions. For the current benchmark, we use only the RGB streams to maintain compatibility with standard open-source models and ensure a fair comparison with the web data. Depth information is preserved and included in the dataset release. Although not used in the reported baselines, the depth and multi-view streams are intended to support future work on 3D hand reconstruction and multimodal recognition. We have clarified this in Section Limitation and Future Work of the revised manuscript.
>
> **W3/Q1/Q3/Q4: Clarification of the annotation protocols.**
>
> **Response:** Thank you for your comment. We clarify the annotation protocols as follows. We prioritize linguistic realism in all annotations. Spelling errors in labels are preserved exactly as signed (for example, annotating "MAQUIRE" if the signer misspells "Maguire"), allowing the dataset to reflect real-world noise rather than a sanitized version of fingerspelling. For segmentation, annotators perform precise temporal delineation, marking the exact start and end timestamps of the full fingerspelled sequence within the video. In addition, we annotate "lexical forms", defined as the standard English words corresponding to the fingerspelled sequences. This semantic alignment is crucial for interpreting abbreviations (such as "EQ") and acronyms, and for mapping them back to their intended target meanings (for example, "Equipment").
>
> More detailed annotation instructions are provided in Appendix C.4 ("Fingerspelling Annotation Guidelines").
>
> **W4/Q6: Presentation Error.**
>
> **Response:** Thank you for pointing this out. We have carefully proofread the entire manuscript and corrected all spelling and grammar mistakes in the revised version.
>
> **Q2: Signer Identification.**
>
> **Response:** Thank you for the question. For videos containing multiple individuals, AlphaPose is used to track all people in the scene and generate identity-consistent pose trajectories. Annotators then manually review the overlaid trajectories and explicitly select the signer based on spatial position and continuous signing motion. Thus, signer identity is always human-verified rather than automatically assigned. We have clarified this in Section 3.1 of the revised manuscript.
>
> **Q5: Relation between Hearing Status and Signing Level.**
>
> **Response:** Thank you for the question.
> Hearing status is not used as an indicator of signing proficiency in our dataset. Proficiency is instead reflected by explicit participant categories: "Auslan Experts" and "Deaf individuals" represent native or professional-level signing ability, while "Volunteers" correspond to sign language learners in the lab setting. Researchers can use these category labels to subset the dataset according to proficiency level.

---

> > ### Author Response · Authors · 2025-11-26
> > **Response to Reviewer qoMh (2/2)**
> >
> > **Response to Ethical Concerns.**
> >
> > We take ethical responsibility and legal compliance seriously and provide clarifications below.
> >
> > - Legal compliance for web-sourced data. For YouTube and other online videos, we strictly follow platform Terms of Service. We do not download, re-host, or redistribute any copyrighted audiovisual material. The dataset contains only our annotations and the official public URLs of the original videos, and all playback remains hosted under the creator’s control on YouTube. Dataset users are required to adhere to the corresponding platform ToS. A formal takedown mechanism is provided, allowing rights holders to request immediate removal or anonymization of any entry. These practices follow the same non-redistributive protocols used in prior sign-language datasets such as YouTube-SL-25 [1] and Auslan-Daily [2].
> >
> > - Consent form standards. The data collection protocol and informed consent form used in the Lab Recordings subset (Appendix D) were reviewed and approved by the University’s Research Ethics Committee. The form explicitly authorizes the recording and academic release of identifiable facial and hand data, consistent with standard human-subject research guidelines. A withdrawal and face-blurring protocol is in place to honor participant requests.
> >
> > More details are provided in the Ethics Statement in the revised manuscript.
> >
> > [1] YouTube-SL-25: A Large-Scale, Open-Domain Multilingual Sign Language Parallel Corpus, ICLR 2025.
> >
> > [2] Auslan-Daily: Australian Sign Language Translation for Daily Communication and News, NeurIPS 2023.

---

> > > ### Comment · Reviewer_qoMh · 2025-11-26
> > >
> > > I would like to thank the authors for addressing the weaknesses and questions in my review. Considering the importance of creating new dataset for sign language recognition and the quality of the paper, I will keep my score at 8 (accept).

---

> > > > ### Author Response · Authors · 2025-11-26
> > > > **Official Comment by Authors**
> > > >
> > > > Dear Reviewer qoMh,
> > > >
> > > > Thank you for your positive response and valuable comments! We are glad that our clarifications addressed your concerns.
> > > >
> > > > Best regards,
> > > >
> > > > Authors.

---

### Official Review · Reviewer_YuAS · 2025-10-31

**Soundness:** 4
**Presentation:** 3
**Contribution:** 4
**Rating:** 8
**Confidence:** 4

**Summary:**

The paper introduces BANZ-FS, a new, large-scale dataset focused on two-handed fingerspelling for BANZSL (British, Australian, and New Zealand Sign Language). The authors highlight that existing sign language datasets rarely address the specific challenges of two-handed fingerspelling, focusing instead on single-handed systems like American Sign Language (ASL). The dataset features rich, multi-level annotations, including video-to-subtitle, video-to-letter, and video-to-lexicon alignments. It is designed to capture real-world linguistic phenomena like spelling errors, acronyms, and abbreviations , as well as visual challenges unique to two-handed signing, such as self-occlusion and rapid transitions. The authors benchmark state-of-the-art models on tasks like fingerspelling detection, isolated recognition, and recognition in context, demonstrating that the dataset presents significant challenges for current methods.

**Strengths:**

1. Novel and Specific Contribution: This is the first large-scale fingerspelling dataset specifically for the two-handed BANZSL system. It fills a critical gap, as most resources are for single-handed systems.

2. High Data Diversity: By combining news, lab, and web sources, the dataset captures a broad spectrum of signing styles. This includes formal signing from professionals, clean signing in controlled settings, and casual "in-the-wild" signing from 116 unique signers.

3. Linguistic Realism: A major strength is the inclusion and annotation of real-world linguistic phenomena often missing from clean datasets. This includes spelling errors (e.g., "Maguire" signed as "Maquire"), lexical abbreviations ("EQ" for "equipment"), acronyms ("GWS"), and self-corrections.

4. Rich, Multi-Level Annotations: The dataset provides comprehensive annotations beyond simple transcription. This multi-level alignment supports a variety of tasks, including detection (temporal boundaries), isolated recognition (letter sequences), and contextual recognition (alignment with spoken language lexicons).

5. Robust Benchmarking: The authors establish strong baselines by testing existing models on the new data. The results (Tables 3 & 4) show that current methods struggle with cross-domain generalization (e.g., training on Lab data and testing on Web data), proving the dataset is a challenging and valuable resource for advancing the field.

**Weaknesses:**

1. The methods evaluated in Tab.3 and Tab.4 are a bit old, which are mostly before 2023. For a paper submitted to ICLR 2026, new state-of-the-art methods could be incorporated.
2. As shown in Tab.2, the dataset is "heavily skewed toward Auslan" (Australian Sign Language). The BSL (British) and NZSL (New Zealand) contributions are "relatively limited" , primarily coming from the smaller "Web Videos" and "YouTube" sources.
3. While the proposed dataset highlight the "in-the-wild" characterisitic, this is not fully reflected in the data construction and composition. Most videos are still captured from News or Lab environments.
4. Some labelling steps remain confusing. For example, how to (2) identify the signer ID based on pose trajectories? Besides, only a little data is cross-checked. How to guarantee the data quality is a question.

**Questions:**

See above

---

> ### Author Response · Authors · 2025-11-26
> **Response to Reviewer YuAS**
>
> We sincerely thank **Reviewer YuAS** for the encouraging feedback. We are pleased that the reviewer recognizes BANZ-FS as a novel and specific contribution filling a critical gap in two-handed fingerspelling resources, and appreciates its high data diversity, linguistic realism, and robust benchmarking. We address the constructive comments below.
>
>
> **W1: Evaluation on the latest state-of-the-art methods.**
>
> **R1:** Thank you for the suggestion.
> For the core computer vision tasks, fingerspelling detection and recognition, we benchmark all publicly available state-of-the-art methods, including HandReader (May 2025) [1], the most recent high-accuracy recognizer. Table A summarizes its performance across different training subsets and evaluation domains.
>
> **Table A**: Letter-level accuracy of HandReader [1] across different training subsets and test domains in BANZ-FS
> | **Method**         | **Train Split** | **News** | **Lab** | **Web** | **Full** |
> | ------------------ | --------------- | -------- | ------- | ------- | -------- |
> | **HandReader [1]** | Full            | 64.4     | 86.7    | 71.8    | 75.4     |
> |                    | News            | 68.3     | 55.6    | 48.5    | 60.8     |
> |                    | Lab             | 37.1     | 93.1    | 32.6    | 63.1     |
> |                    | Web             | 29.8     | 48.1    | 55.0    | 40.2     |
>
>
> Although HandReader achieves a strong baseline (75.4\% on Full), significant domain gaps persist. We have integrated these results into the revised version (Section 5.2) and continue to update the benchmark as new methods emerge.
>
> [1] HandReader: Advanced Techniques for Efficient Fingerspelling Recognition. arXiv preprint 2025 (15 May).
>
>
> **W2: Clarification on dialect imbalance.**
>
> **R2:** Thank you for your comment. While Auslan constitutes the majority of the dataset, we emphasize that all three BANZSL dialects (Auslan, BSL, NZSL) share an identical two-handed manual alphabet, meaning the core visual features for fingerspelling are linguistically consistent.
> As shown in Tables 3 and 4 of the main paper, models trained on the Auslan-heavy subsets demonstrate effective generalization to the Web subset (which contains BSL/NZSL data).
> Furthermore, as explicitly noted in the LIMITATION AND FUTURE WORK section, we have planned to use our trained BANZ-FS detector to mine additional BSL/NZSL clips from broadcast sources to address this imbalance in future expanded releases.
>
> **W3: Clarification on "in-the-wild" characteristics.**
>
> **R3:** Thank you for your comment. We clarify that we explicitly introduced the Web subset to capture "in-the-wild" visual characteristics (Figure 1) specifically to complement the environmentally controlled News and Lab settings. While News and Lab data currently comprise the majority to ensure high-quality alignment for training, the Web subset serves as a critical testbed for evaluating model robustness in unconstrained scenarios.
> We acknowledge the need for a more balanced composition and are committed to collecting more "in-the-wild" web videos in our future work to further enhance this dimension.
>
>
> **W4: Clarification on labelling steps and quality control.**
>
> **R4:** Thank you for the helpful comments. We provide detailed clarification below.
>
> - Signer identification. For videos containing multiple individuals, AlphaPose is used to track all people in the scene and generate identity-consistent pose trajectories. Annotators then manually review the overlaid trajectories and explicitly select the signer based on spatial position and continuous signing motion. Thus, signer identity is always expert-confirmed rather than automatically assigned.
>
> - Annotation reliability. To ensure label quality, we employ a recognition-based verification protocol. Each annotator (examiner) cross-checks a random five percent sample of clips annotated by another expert. The examiner independently recognizes the fingerspelling content within the annotated temporal boundaries. A sample is accepted only when the recognition matches the annotated label. In practice, approximately 95% percent of sampled batches passed verification in the first round. Following our protocol, if more than ten percent of samples in a batch contain errors, the entire batch is rejected and a third expert performs adjudication.
>
> We have incorporated these clarifications into the revised manuscript, and the updated description in Section 3.1.

---

### Official Review · Reviewer_DWDk · 2025-11-03

**Soundness:** 2
**Presentation:** 3
**Contribution:** 3
**Rating:** 4
**Confidence:** 4

**Summary:**

This paper introduces BANZ-FS, a dataset for two-handed BANZSL fingerspelling, combining ABC News Auslan broadcasts, lab multi-camera RGB-D recordings, and web-sourced videos. The dataset contains >35k annotated fingerspelling segments from 116 signers, with multi-level alignments (video ↔ subtitles ↔ letters ↔ lexicon). The authors define and benchmark four tasks (IFSR, FSD, FSD-R, FSR-Context), evaluate pose- and RGB-based baselines, present cross-domain experiments, and report qualitative case studies.

**Overall:** the dataset is useful and fills a real gap; however, the manuscript needs clearer documentation of procedures, metrics, and ethics before the benchmark claims can be fully trusted.

**Strengths:**

**Presentation:** The paper is clearly structured (Intro → Related Work → Data Collection → Statistics → Tasks → Benchmarks → Case Studies). Tables summarizing dataset comparisons and statistics (Table 1 & Table 2) are informative and well-labeled. The taxonomy of tasks (IFSR, FSD, FSD-R, FSR-Context) is clearly defined and consistent throughout.

**Contribution:** The proposed BANZ-FS dataset addresses a tangible linguistic and practical gap by focusing on two-handed BANZSL fingerspelling, a setting largely underrepresented in current literature. Its annotation depth (multi-level temporal alignment, OOFS/singleton reporting, and FS speed distribution) and multi-domain composition (news, lab multi-view, web) provide meaningful diversity and reusability for the community.

**Techniques:**

1. A well-designed multi-domain acquisition strategy combining professional broadcast data, lab-controlled RGB-D recordings, and in-the-wild web footage, enabling both realism and controlled benchmarking.

2. Rich, fine-grained annotations supported by comprehensive statistics (e.g., long-tail letter frequency, signer diversity, OOFS/singletons, FS speed).

3. A complete benchmark suite spanning isolated recognition, detection, joint detection-recognition, and contextual evaluation, covering both constrained and realistic use cases.

4. Practical qualitative case studies that transparently illustrate typical failure modes such as rapid fingerspelling or loose boundary segmentation, adding diagnostic insight to the quantitative results.

**Weaknesses:**

**1. Segmentation design lacks empirical justification:** Section 3.3 adopts a fixed 10s sliding window without ablation or rationale. An analysis of alternative window lengths (e.g., 5s, adaptive) is needed to confirm robustness.


**2. Annotation reliability not quantified:** Although multi-level verification is mentioned (Section 3.1), no inter-annotator agreement (e.g., κ, α) or adjudication detail is provided. This weakens confidence in temporal boundary quality.


**3. Evaluation criteria under-specified:** Metrics such as AP@IoU and AP@Acc (Section 5.2) lack exact definitions and sensitivity checks. Clear reporting of thresholds and pipeline details is essential for reproducibility.


**4. Cross-domain variability insufficiently analyzed:** Results in Table 3 - 4 reveal strong domain gaps, yet no normalization or domain ablation is provided. A domain-wise feature analysis (e.g., t-SNE) would clarify generalization behavior.


**5. Error analysis remains anecdotal:** Case studies (Section 5.3) illustrate failures qualitatively but lack quantitative breakdowns, e.g., per-letter confusion or temporal precision - which would turn examples into diagnostic insight.

**6. Signer-split clarity:** Section 3.3 does not specify whether the splits are signer-disjoint. Please clarify whether signer identities overlap across splits, and report this information explicitly.

**7. Ethical transparency missing:** No explicit consent, license, or release protocol is reported, though videos include identifiable individuals. An ethics statement is required for dataset release.

**Questions:**

Most of my concerns and suggestions are already reflected in the Weaknesses section. I believe that clarifications or additional experiments addressing these issues would substantially strengthen the paper.

**Details Of Ethics Concerns:**

**Concerns:** privacy and consent for identifiable video data, annotator treatment, and legal compliance for scraped broadcast/YouTube content.

---

> ### Author Response · Authors · 2025-11-26
> **Response to Reviewer DWDk (1/2)**
>
> We sincerely thank **Reviewer DWDk** for the constructive feedback. We are encouraged that the reviewer recognizes the value of BANZ-FS in filling a tangible linguistic and practical gap for the underrepresented two-handed fingerspelling task. We also appreciate the recognition of our well-designed multi-domain acquisition strategy and rich, fine-grained annotations.
> We respond to the questions raised by the reviewer below.
>
> **W1: Segmentation design lacks empirical justification.**
>
> **R1:** Thank you for the constructive comment. We adopted a fixed 10s sliding window for the following reasons:
>
> - Robustness via Temporal Randomness: As described in Section 3.3, we apply a "10-second sliding window around any detected FS segment". Our design allows the fingerspelling segment to appear at variable temporal positions within the window rather than being strictly centered. This introduced randomness serves as a form of data augmentation, encouraging the model to learn temporal invariance and enhancing its robustness.
> - Coverage of Long Sequences: While Figure 3(b) indicates that many isolated segments are short (<1.5), Table 8 in the Appendix reveals that continuous fingerspelling clips (particularly from YouTube) have an average length of 6.25 seconds, with some significantly exceeding this duration. A shorter window would pose a risk of truncating these longer, naturalistic sequences.
> - Ablation Study (5s vs 10s): Following your suggestion, we re-evaluated the Fingerspelling Detection (FSD) task on the test set using a 5-second sliding window. As shown in Table A, we found that the performance of the baseline model remained stable with negligible differences compared to the 10-second setting. This stability indicates that our training strategy effectively handles the temporal context provided by the larger window.
>
> **Table A**: Ablation study on sliding window size for Fingerspelling Detection (AP@IoU0.5)
> | **Window Size** | **News** | **Lab** | **Web** | **Full** |
> | --------------- | -------- | ------- | ------- | -------- |
> | 5 seconds       | 53.4     | 82.7    | 45.9    | 66.2     |
> | 10 seconds      | 53.9     | 82.7    | 47.3    | 66.9     |
>
> We have included these justifications and experiments in Appendix Section H.1.
>
>
> **W2: Annotation reliability not quantified.**
>
> **R2:** Thank you for the comment. We ensured data reliability through a rigorous Expert Cross-Check and Adjudication pipeline. Specifically, our adjudication process employed a "recognition-based verification" method: a second expert (examiner) independently recognized the fingerspelling content within the temporal boundaries defined by the initial annotator.
> If the recognition matched the label, the annotation was deemed valid.
>
> In practice, we observed high initial consistency, with approximately 95% of the sampled batches passing the strict verification threshold (error rate <10%) in the first round without requiring re-annotation.
> This high pass rate serves as a robust quantitative proxy for the reliability of our temporal boundaries and labels.
> We have included these specific statistics and the detailed adjudication workflow in Section 3.1 of the revised manuscript.
>
> **W3: Evaluation criteria under-specified.**
>
> **R3:** Thank you for the question.
> As outlined in Section 4 and the caption of Table 4, our evaluation metrics and thresholds are strictly consistent with established protocols in previous work [1] to ensure fair comparison.
> Specifically:
> - AP@IoU: Represents Average Precision at a temporal Intersection-over-Union (IoU) threshold of 0.5, denoted as AP@IoU0.5.
> - AP@Acc: Represents Average Precision where a detection is considered a True Positive if the downstream recognition accuracy exceeds 50%, denoted as AP@Acc0.5.
> Following your suggestion, we explicitly specify thresholds in the revised Section 4.
>
> [1] Fingerspelling Detection in American Sign Language. CVPR 2021.
>
> **W4: Cross-domain variability insufficiently analyzed.**
>
> **R4:** Thank you for your suggestion. We respectfully point out that Table 3 already serves as a quantitative domain ablation study. By training on individual subsets (News, Lab, Web) and testing on others, we explicitly quantified the generalization gap. For instance, HandReader [2] trained on News drops from 68.3% accuracy on News to 48.5% on Web.
>
> To further clarify the generalization behavior as requested, we have included a t-SNE visualization of RGB features [2] in Appendix Section H.2 of the revised manuscript.
> The visualization visually illustrates the distributional shifts between the controlled Lab environment, the professional News studio, and the unconstrained Web settings.
>
> [2] HandReader: Advanced Techniques for Efficient Fingerspelling Recognition. arXiv preprint 2025 (15 May).

---

> > ### Author Response · Authors · 2025-11-26
> > **Response to Reviewer DWDk (2/2)**
> >
> > **W5: Error analysis remains anecdotal.**
> >
> > **R5:** Thank you for the constructive suggestion. While Section 5.3 provided qualitative case studies, we have included a quantitative analysis in Appendix Section H.3 of the revised manuscript.
> > Specifically, we provide a **Per-Letter Confusion Matrix** to verify sources of visual ambiguity, and a **Performance vs. Speed Analysis** to empirically quantify the performance degradation associated with increased signing tempo.
> > These additions transform our anecdotal observations into concrete diagnostic insights.
> >
> > **W6: Signer-split clarity.**
> >
> > **R6:** Thank you for the question regarding signer splits. We clarify that the splits are not signer-disjoint and signer identities do overlap across splits. To ensure transparency, we have conducted a detailed census of signer distribution. We identified OOS (out-of-training Signers) as signers that never appear in the training set to gauge generalization to unseen users.
> > As shown in Table B, the Test sets for News, Lab, and Web contain 5, 12, and 2 OOS signers respectively. We have included these details in Section 3.3.
> >
> > Table B: Signer distribution across data sources and splits. OOS indicates signers that never appear in the training set.
> > | **Data Source**    | **Split**   | **Train** | **Valid** | **Test** |
> > | ------------------ | ----------- | --------- | --------- | -------- |
> > | **ABC News**       | Num. Signer | 24        | 22        | 19       |
> > |                    | Num. OOS    | 0         | 5         | 5        |
> > | **Lab Recordings** | Num. Signer | 65        | 52        | 50       |
> > |                    | Num. OOS    | 0         | 10        | 12       |
> > | **YouTube**        | Num. Signer | 18        | 10        | 7        |
> > |                    | Num. OOS    | 0         | 3         | 2        |
> >
> >
> >
> > **W7: Ethical transparency missing.**
> >
> > **R7:** Thank you for highlighting these important ethical considerations.
> > We have added explicit clarification in the **ETHICS STATEMENT** and **Appendix Section D** to address privacy, consent, annotator treatment, and the handling of online data sources.
> >
> > **Privacy \& Consent.** All Lab participants signed a detailed consent form (Figure 8) authorizing the recording and academic release of their facial expressions and hand movements. We maintain a formal withdrawal protocol, including immediate data removal or face-blurring upon request.
> >
> > **Legal compliance for online content.** For broadcast and YouTube videos, we strictly comply with platform Terms of Service: we do not download, re-host, or redistribute any copyrighted audiovisual material. The dataset release contains only our annotations and the official public URLs, and all video playback remains hosted under the creator’s control. A takedown mechanism is provided for content creators who wish their material to be removed.
> >
> > **Annotator treatment.** All annotators were compensated at fair professional rates: general volunteers at AUD \\$40/hour and Deaf signers/Auslan consultants at AUD \\$100/hour, in line with institutional standards.
> >
> > Pointers have been added in Sections 3.1 and 3.2 directing readers to the ETHICS STATEMENT for full details. We hope this clarifies that BANZ-FS is accompanied by transparent and comprehensive ethical safeguards.

---

> > ### Comment · Reviewer_DWDk · 2025-11-27
> >
> > Thank you for the responses, all of my concerns are addressed. I also really appreciate the extra experiments the authors added. With everything clarified, I will accordingly raise my score.

---

> > > ### Author Response · Authors · 2025-11-27
> > > **Official Comment by Authors**
> > >
> > > Dear Reviewer DWDk,
> > >
> > > Thank you for your positive response and valuable comments! We are glad that our clarifications addressed your concerns.
> > >
> > > Best regards,
> > >
> > > Authors.

---

### Meta-Review · Area_Chair_CxRv · 2025-12-08

**Summary:**

The authors introduce a novel large-scale data set for two-handed fingerspelling for BANZSL (British, Australian, and New Zealand Sign Language). The reviewers highlight the importance of the contribution and raised many detailed questions that were clarified in the rebuttal. The current version of the manuscript contains all revisions.

**Reviewer Concerns:**

All concerns have been successfully addressed by the authors.

**Reviewer Scores:**

Several reviewers had been convinced by the original submission already and some few others offered to raise their score after the rebuttal. The authors did a good job in providing requested information and clear up incaccuracies and/or misunderstandings.

---

### Decision · Program_Chairs · 2026-01-26

Accept (Poster)